# MOTOR: A Time-to-Event Foundation Model for Structured Medical Records

**Ethan Steinberg**[1][*]**, Jason Alan Fries**[2][*]**, Yizhe Xu**[2]**, Nigam H. Shah**[3]

[1]Department of Computer Science, Stanford University
[2]Center for Biomedical Informatics Research, Stanford University
[3]Stanford University, Stanford Health Care
`{ethanid, jfries, yizhex, nigam}@stanford.edu`

## ABSTRACT

We present a self-supervised, time-to-event (TTE) foundation model called MOTOR (Many Outcome Time Oriented Representations) which is pretrained on timestamped sequences of events in electronic health records (EHR) and health insurance claims. TTE models are used for estimating the probability distribution of the time until a specific event occurs, which is an important task in medical settings. TTE models provide many advantages over classification using fixed time horizons, including naturally handling censored observations, but are challenging to train with limited labeled data. MOTOR addresses this challenge by pretraining on up to 55M patient records (9B clinical events). We evaluate MOTOR's transfer learning performance on 19 tasks, across 3 patient databases (a private EHR system, MIMIC-IV, and Merative claims data). Task-specific models adapted from MOTOR improve time-dependent C statistics by 4.6% over state-of-the-art, improve label efficiency by up to 95%, and are more robust to temporal distributional shifts. We further evaluate cross-site portability by adapting our MOTOR foundation model for six prediction tasks on the MIMIC-IV dataset, where it outperforms all baselines. MOTOR is the first foundation model for medical TTE predictions and we release a 143M parameter pretrained model for research use at https://huggingface.co/StanfordShahLab/motor-t-base.

## 1 INTRODUCTION

Many applications in medicine require predicting not only the overall likelihood of an event, but also when that event will occur. *Time-to-event (TTE) models* (also called *survival models*) are commonly used to estimate the probability distribution of TTE data given a set of features. TTE models have two primary advantages over standard classification approaches using fixed time horizons. First, they can estimate temporal quantities of interest such as the hazard rate or the median time until an event occurs. Second, TTE models account for various types of censoring induced by different data collection mechanisms, which is critical for obtaining accurate estimates and fair models (Chang et al., 2022). TTE models are commonly used, *inter alia*, in medicine (Lambert et al., 2010; Lin et al., 2014; Fernández et al., 2016; Alaa & van der Schaar, 2017), churn prediction (Perianez et al., 2016), real time advertisement bidding (Wu et al., 2015), recommendation systems (Jing & Smola, 2017), and other business analytics.

Despite the benefits of TTE models, they come with the downside of requiring large training sets in order to estimate TTE probability distributions without bias (Wey et al., 2015). High censoring rates, common in medical events with long time horizons, further limit the available training data and exacerbate the risk of

---

[*]The authors contributed equally to this paper.

overfitting. A variety of methods have been proposed to mitigate this issue, generally by making assumptions about the TTE distribution such as proportional hazards (Cox, 1972) or parametric distributions (Martinsson, 2016; Avati et al., 2020). These assumptions significantly reduce the number of learnable parameters, but may also degrade predictive performance when they do not hold (Schemper, 1992).

Recent work in deep learning has approached the challenge of limited training data by relying on transfer learning via increasingly powerful *foundation models* (Bommasani et al., 2021). Foundation models leverage advances in self-supervised learning to train models using large collections of unlabeled data. The resulting pretrained models can be quickly adapted to new tasks while improving accuracy, using less labeled training data, and being more robust to distributional shifts (Liang et al., 2023) – key desiderata in medical machine learning. Prior research has explored self-supervised pretraining with structured medical records (procedure and diagnosis codes, lab results, etc.) by using autoregressive or "next code"-style pretraining objectives (Steinberg et al., 2021), however significant gaps remain when considering TTE modeling. First, large-scale pretraining using heterogeneous patient data is underexplored for TTE modeling, with prior work focusing on models trained from scratch, using limited input features, small numbers of target tasks, and disease-specific patient cohorts (Li et al., 2016; Wang & Sun, 2022). Second, standard autoregressive pretraining objectives may not be effective for TTE modeling, as they do not directly capture long-term time horizons commonly found in patient medical records. Finally, existing pretrained foundation models for structured medical record data are generally not released to researchers (Wornow et al., 2023), creating serious reproducibility challenges when evaluating transfer learning, the primary benefit afforded by foundation models.

To address these challenges, we introduce MOTOR (Many Outcome Time Oriented Representations), a foundation model for medical TTE trained on up to 55M patient records (comprising 9B clinical events). Our main contributions are:

1. A self-supervised pretraining objective that uses thousands of TTE tasks (8,192 in our experiments) during pretraining to learn broadly useful features for TTE modeling. This novel pretraining task outperforms autoregressive pretraining by an average of 2.0%.

2. An efficient Transformer-based piecewise exponential model architecture that leverages shared time-dependent weights and sparsity to scale to thousands of TTE pretraining tasks while reducing memory usage by up to 35%.

3. Experiments demonstrating that TTE pretraining improves accuracy by an average of 4.6% over state-of-the-art methods for 19 evaluation tasks. We further verify that MOTOR has many important foundation model properties, including robustness over time, the ability to transfer a pretrained model for reuse in another dataset, and the ability to fine-tune task models with limited labeled data.

4. A 143M parameter MOTOR model pretrained on EHR data, released for non-commercial research use. To our knowledge, this is the first release of a medical foundation model for TTE.

## 2  RELATED WORK

**Time-to-Event Models:** Accelerated failure time (AFT) models are a fully parametric approach for modeling TTE outcomes where the TTE distribution is parameterized with a specified probability distribution. AFT models assume that the parameters of that specified probability distribution are a learnable function of the features. Common probability distributions include exponential, gamma, log-normal, log-logistic, and Weibull. Piecewise distributions enable the use of different distribution parameters across different regions of time and thus provide additional flexibility, with the piecewise exponential distribution (Friedman, 1982) being a popular choice. More recently, AFT models have been combined with deep learning by using a neural network to map between input features and distribution parameters (Fornili et al., 2014; Martinsson, 2016; Ren et al., 2019; Avati et al., 2020; Ryu et al., 2020; Lee et al., 2020; Kopper et al., 2021; Kvamme & Borgan, 2021; Nagpal et al., 2021).

Cox proportional hazards models (PH) (Cox, 1972) reduce the complexity of the model space by assuming a constant hazard ratio between instances. Cox models can also be combined with deep learning by using a neural network to map between the input features and the learned hazard ratio (Katzman et al., 2018). Random survival forests (RSF) (Ishwaran et al., 2008) are an extension of random forests to the time-to-event setting. The key features of RSF are that it supports non-linear feature interactions and is non-parametric.

Some work has explored using multi-task learning to fit TTE models with limited data. Li et al. (2016) proposed treating the pieces in piecewise TTE models as instances of related tasks and regularizing models to share weights across pieces. Gao & Cui (2022) explored choosing "compatible" clinical tasks (i.e., clinical tasks with similar risk structure) to better learn neural network featurized Cox models. Wang & Sun (2022) evaluated learning mortality and length of stay models simultaneously while training cancer models. Lim & van der Schaar (2018) combined a TTE task with two continuous and twenty binary tasks to learn better models for cystic fibrosis. Kim et al. (2020) used existing VAE techniques to pretrain cancer survival models.

MOTOR differs from these approaches in several ways. First, we focus on a self-supervised pretraining objective that captures temporal structure from clinical events. Second, we massively increase the scale of patients (up to 55M) and tasks (8,192 tasks) used for pretraining and outline methods required to efficiently train at this scale. Finally, we explore the benefits of this pretraining on various downstream adaptation tasks.

**Deep Learning for Structured Medical Record Data:** There is considerable prior work using deep learning with structured medical records for medical classification tasks. State-of-the-art work directly incorporates the temporal structure of medical record data by using recurrent neural networks (RNNs) (Che et al., 2016; Choi et al., 2016) or Transformers (Li et al., 2020). While initial deep learning efforts focused on end-to-end training using small, task-specific cohorts, more recent work has explored using self-supervised learning to pretrain models using millions of patient records. Various self-supervised objectives have been proposed for learning meaningful patterns in structured medical record data, including denoising autoencoders (Miotto et al., 2016), autoregressive "next day" code prediction (Steinberg et al., 2021), and masked language modeling (Li et al., 2020; Rasmy et al., 2021b; Pang et al., 2021; Zeng et al., 2022).

The majority of recent architectures are formulated for classification tasks with fixed time horizons, e.g., 3-12 month mortality (Avati et al., 2018). However, there are several efforts that combine deep learning with TTE modeling on EHR data. Avati et al. (2020) used RNNs and feedforward neural networks to power AFT models for estimating time until death. Rasmy et al. (2021a) used RNNs to train Cox models for hospital mortality and ventilation need. Wen et al. (2022) used feedforward networks to parameterize Cox and DeepHit models for predicting time until discharge. MOTOR differs from these prior works in two ways: 1) We use a Transformer-based piecewise exponential model to handle complex and nonlinear relationships in data, and 2) we leverage TTE pretraining at a massive scale to estimate model parameters more effectively.

## 3 METHODS

**Problem Setup:** We consider a set of $N$ patients and their medical records $X$, where $X_i$ denotes patient $i$'s information and consists of a sequence of timestamped events, where each event is defined by a code that is a symbol drawn from a finite set defined by standard medical ontologies (Sciences & Informatics, 2021). Let $X_{ij}$ denote the $j$-th event in the sequence, where each event consists of a code for that event and the corresponding timestamp for when that event occurs.

We are interested in learning models to predict the time-to-event for various tasks. Let $T_i$ denote the event time and let $C_i$ denote the censoring time for patient $i$. Our goal is to estimate the probability distribution of event times $\mathbb{P}(T_i = t)$, accounting for censoring. We make the standard non-informative censoring assumption (i.e. that censoring times are independent of event times).

**MOTOR Overview:** We introduce MOTOR, a Transformer-based neural network model that is pretrained with a self-supervised, TTE objective defined using structured medical records data. Figure 1 provides an overview of our approach, where inputs are patient timelines of time stamped clinical events. Once pretrained, MOTOR is combined with a linear head to train downstream models for particular tasks of interest such as predicting the time until a heart attack.

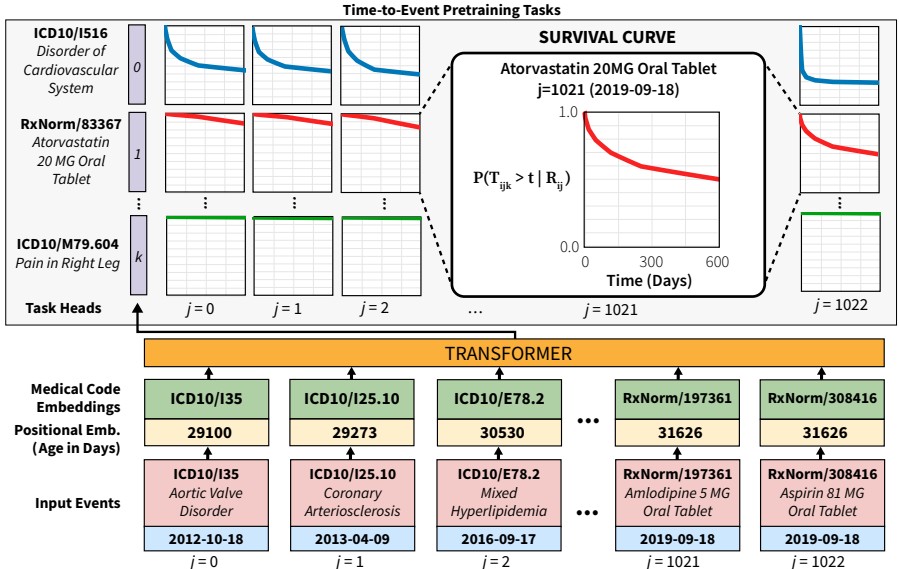

Figure 1: MOTOR architecture and TTE pretraining tasks, as applied to a patient. Note how tasks (survival curves) vary at each time step $j$, e.g., atorvastatin risk increases given a history of cardiovascular disorders.

**Transformer:** We use a standard Transformer with local attention and causal masking (Roy et al., 2021) to compute embeddings for every patient at every time step. The full details for how this is implemented to process structured medical data, including time embeddings, is specified in Appendix F, but the overall theme is to transform each patient $X_i$ into a single matrix per patient $R_i$, where each row $R_{ij}$ corresponds to patient $i$'s representation for event $j$ that contains cumulative information up to and including event $j$.

**Piecewise Exponential Time-to-Event Model:** Given the representations computed by the Transformer, MOTOR constructs a large-scale TTE model that can generate survival predictions for thousands of tasks. We derive and implement a scalable approach based on the Piecewise Exponential Artificial Neural Networks (PEANN) (Fornili et al., 2014) framework that can work with a large number of tasks.

Piecewise exponential models approximate the TTE distribution with a piecewise exponential function using a total number of pieces, $P$. Each piece $p$ covers the time period starting at $S_p$ and ending at $E_p$. For every patient $i$, event $j$, task $k$, and piece $p$, a hazard $\lambda_{ijkp}$ must be estimated to formulate a piecewise exponential TTE distribution for each patient. Formally, the piecewise exponential distribution is defined by the hazard function $H_{ijk}(t) = \sum_{p=1}^{P} I(S_p \leq t < E_p)\lambda_{ijk}$ and the survival function $S_{ijk}(t) = \prod_{p=1}^{P} e^{-\lambda_{ijkp}*(min(t,E_p)-S_p)*I(t \geq S_p)}$.

This piecewise exponential model is combined with the event indicator tensor $\delta$ (defined as $\delta_{ijkp}$ indicating whether or not subject $i$ at the time of event $j$ has an event for task $k$ within the piece $p$) and event time tensor $U$ (defined as $U_{ijkp}$ being the time to either the task event or censorship within piece $p$ for patient $i$, event $j$

and task $k$) to construct the standard piecewise exponential likelihood loss function:

$$L(U_{ijkp}|\lambda_{ijkp}) = \{\lambda_{ijkp}\exp(-\lambda_{ijkp}\,U_{ijkp})\}^{\delta_{ijkp}}\{\exp(-\lambda_{ijkp}\,U_{ijkp})\}^{1-\delta_{ijkp}} \qquad (1)$$

**Large Scale Piecewise Exponential Optimizations:** Computing this loss requires the evaluation of large matrix operations due to the dependence on $i$, $j$, $k$, and $p$, so we employ three optimizations to make this more tractable. First, we assume that the transformation $R_{ij} \mapsto \log(\lambda_{ijkp})$ is a linear mapping of low rank, and compute it with two smaller matrix multiplications. This is done by learning a time-independent task embedding $\beta_k$ of size $b$ for each pretraining task $k$ and learning a time-dependent linear transformation to $M_{ijp}$ from patient representation $R_{ij}$, in which $M_{ijp}$ represents the state of patient $i$ at event $j$ within piece $p$. We can then compute $\log(\lambda_{ijkp}) = M_{ijp} \cdot \hat{\beta}_k$. This requires $P$ times fewer parameters per task compared to a full rank transformation from $R_{ij}$ to $\lambda_{ijkp}$. The intuition here is that tasks should be largely time-independent as the risk factors for diseases mostly remain constant. Tasks further exhibit a high degree of censorship, with a low probability for any particular event in a time bin, meaning $\delta$ and $U$ can be implemented using sparse matrix operations to significantly reduce compute and memory. For example, in our STARR-OMOP experiments, only 0.6% of entries are non-duplicate, reducing the memory required to store and process $\delta$ and $U$ from 4 GB to 0.05 GB per batch. Finally, as we only require the product of the likelihood loss function and not the individual terms, we fuse most operations into a single kernel such that large matrix intermediates like $\lambda$ and $\exp(-\lambda_{ijkp}\,U_{ijkp})\}$ are never fully materialized. This reduces memory by a further 4 GB per batch. As our largest trained architecture (12 layers) itself requires 14.59 GB of memory, these optimizations combine to reduce our memory requirements by 35%.

**Time-to-Event Pretraining:** Given our Transformer encoder and piecewise exponential TTE setup, we define a pretraining task of predicting the time to the next time a code is assigned for various codes. To achieve a general TTE model, we train on a larger number of codes. We use a simple entropy based-heuristic to rank and select the most informative codes, described in Appendix N.

To provide a more conservative measure of performance on novel TTE tasks, we explicitly removed codes for our evaluation target tasks from the set of pretraining tasks. Note that removing codes for evaluation is not strictly necessary for correctness as we are using a patient split to ensure no leakage between pretraining and evaluation. We pretrain our model end-to-end using the likelihood in Equation (1). Hyperparameters are set through a grid search (detailed in Appendix A) on the validation set.

**Adapting MOTOR to Target Tasks:** Given the pretrained model above, we can then train models for new target tasks of interest through the process of adaptation. Specifically, we need to estimate a new vector of parameters $\hat{\beta}_k^{\text{new}}$ which will be used to provide piecewise exponential estimates for the new task. We evaluate two methods of adapting MOTOR, using a linear probe (MOTOR-Probe) and full finetuning (MOTOR-Finetune). In the linear probe setup, only $\hat{\beta}_k^{\text{new}}$ are tuned for the new task and no other weights are modified. This simple approach allows models to be trained quickly as gradients only have to be evaluated over a small fraction of the model. Alternatively, in the full finetuning setup, all of MOTOR's weights are updated when finetuning. This is a flexible approach that can enable higher prediction accuracy but comes at the cost of requiring increased runtime as gradients have to be computed throughout the entire model. As an ablation, we also evaluate training MOTOR from scratch, without any adaptation (MOTOR-Scratch). In this setup, we randomly initialize MOTOR and train the whole model on the target task.

## 4 EXPERIMENTS

We evaluate the effectiveness of MOTOR by comparing the performance of MOTOR-derived models to several state-of-the-art TTE models. All training and evalation code is open source and available on GitHub[1].

---

[1] Code Repository: https://github.com/som-shahlab/motor_code_release

**Datasets and Compute** We use two healthcare datasets for our main experiments: STARR-OMOP, a de-identified EHR dataset from Stanford Hospital (Callahan et al., 2023), and MERATIVE, the 2017 release of the de-identified Merative™ MarketScan® Commercial Claims and Encounters Database (Merative). STARR-OMOP contains EHR data for adult and pediatric populations with a total of 2.7M patient records (2.4B clinical events) spanning 2014 - 2022. Each record includes demographics (age, sex, ethnicity) and medical codes for diagnoses, labs, medications, procedures, and visits. MERATIVE contains claims information (i.e., billing diagnoses, medications, procedures, and visits) for 78M patients (12.8B clinical events) spanning 2007 - 2017. Data processing and splitting details are in Appendix B, with compute information in Appendix C.

**Setup of Target Tasks:** We derive 19 target tasks for evaluating models. Our evaluation set consists of six code-based target tasks (predicting the time to a particular diagnosis code is assigned) and 13 text-based target tasks (predicting the time to a radiology note that refers to a condition). For all tasks, a prediction time is defined by choosing the end of a random visit after at least a year of medical history. We use one sample per patient to better match the structure of existing cohort datasets frequently used for evaluating TTE models (such as SUPPORT or SEER) and avoid issues related to baselines overfitting on highly correlated samples from the same patient. For simplicity, we assume death is the only competing risk and censor each record at death. Since our patient datasets are large, we subsample low-information censored cases (see Appendix D).

The six code-based tasks are Celiac Disease, Heart Attack (HA), Lupus, Nonalcoholic Fatty Liver Disease (NAFLD), Pancreatic Cancer, and Stroke. These tasks were chosen to cover a spectrum of acute and chronic health conditions where a prediction model is believed to have clinical utility (Hujoel et al., 2018; Kingsmore & Lipsky, 2022; Lee et al., 2022; Wong et al., 2022; Razmpour et al., 2023). These clinical events are defined by a set of ICD-10 codes that are expanded using OMOP SNOMED based ontology mappings (Sciences & Informatics, 2021). We remove these codes and related concepts from pretraining to better evaluate performance on unseen tasks. Table S4 in Appendix E lists the codes-based task definitions and statistics.

One limitation of code-based tasks is that coded data may be incomplete and thus many useful tasks can only be defined using information derived from text (Xu et al., 2011). To evaluate this setting, we additionally perform a separate analysis on STARR-OMOP with 13 text-based tasks, using the 13 radiology note NLP labeling functions provided by Irvin et al. (2019). Further details for this evaluation are in Appendix J.4.

**Baselines:** We compare MOTOR-derived models with five other TTE approaches: Cox PH, DeepSurv, DeepHit, DSM, and RSF. Appendix G contains details for how we implemented these baselines.

**Evaluation Metrics:** We evaluate model performance using four metrics, the time-dependent C statistic (Heagerty & Zheng, 2005), Harrell's C-index (Harrell et al., 1982), the Nam-D'Agostino (ND) calibration metric (D'Agostino & Nam, 2003), and the Integrated Brier Score (IBS) (Graf et al., 1999). The C statistics provide an estimate of ranking performance, i.e., the ability of the model to rank higher-risk patients over lower-risk patients. IBS and ND calibration inform us about whether the estimated survival curves contain correct probabilities. Details on these metrics (including statistical significance), are in Appendix H.

## 5 RESULTS

**Overall Performance:** Table 1 provides the time-dependent concordance performance of the various methods for the code-based tasks. Both pretrained MOTOR variants (MOTOR-Finetune and MOTOR-Probe) generally outperform all of our baselines, providing statistically significant higher C statistics in all six code-based tasks (confidence intervals are shown in Table S5 in Appendix I). Comprehensive calibration metrics are also included in the Appendix, specifically Harrell's C-index (J.2) and Integrated Brier Score (J.3). Results on the 13 text-based tasks are in Appendix J.4, with the same pattern of MOTOR outperforming all baselines.

The comparison between MOTOR-Scratch (where the model is trained from scratch only on the target task without any pretraining) and MOTOR-Finetune is particularly important, as it directly tests the importance of

pretraining. As MOTOR combines a sophisticated neural network model with pretraining, it is important to evaluate whether pretraining itself provides value. For a fair comparison, MOTOR-Scratch uses the same piecewise exponential setup, Transformer backbone, and hyperparameter grid as MOTOR-Finetue. MOTOR-Scratch performs worse, demonstrating the importance of pretraining.

Table 1: Time-dependent C statistic on the code-based tasks. Bolding indicates the best performing model.

| Method | Dataset | Celiac | HA | Lupus | NAFLD | Cancer | Stroke |
|---|---|---|---|---|---|---|---|
| Cox PH | STARR-OMOP | 0.689 | 0.761 | 0.770 | 0.726 | 0.793 | 0.779 |
| DeepSurv | - | 0.704 | 0.823 | 0.790 | 0.800 | 0.811 | 0.830 |
| DSM | - | 0.707 | 0.828 | 0.784 | 0.805 | 0.809 | 0.835 |
| DeepHit | - | 0.695 | 0.826 | 0.807 | 0.805 | 0.809 | 0.833 |
| RSF | - | 0.729 | 0.836 | 0.787 | 0.802 | 0.824 | 0.840 |
| MOTOR-Scratch | - | 0.696 | 0.795 | 0.803 | 0.821 | 0.777 | 0.831 |
| MOTOR-Probe | - | 0.802 | 0.884 | 0.850 | 0.859 | 0.865 | 0.874 |
| MOTOR-Finetune | - | **0.802** | **0.887** | **0.863** | **0.864** | **0.865** | **0.875** |
| Cox PH | MERATIVE | 0.538 | 0.783 | 0.749 | 0.799 | 0.628 | 0.693 |
| DeepSurv | - | 0.719 | 0.814 | 0.809 | 0.828 | 0.801 | 0.753 |
| DSM | - | 0.725 | 0.814 | 0.812 | 0.833 | 0.805 | 0.758 |
| DeepHit | - | 0.722 | 0.815 | 0.809 | 0.828 | 0.802 | 0.753 |
| RSF | - | 0.705 | 0.810 | 0.805 | 0.838 | 0.798 | 0.746 |
| MOTOR-Scratch | - | 0.737 | 0.821 | 0.826 | 0.850 | 0.821 | 0.775 |
| MOTOR-Probe | - | 0.755 | 0.828 | 0.833 | 0.856 | 0.825 | 0.789 |
| MOTOR-Finetune | - | **0.762** | **0.831** | **0.838** | **0.862** | **0.834** | **0.794** |

**Choice of Pretraining Objective:** One major part of MOTOR is our TTE pretraining objective. To investigate how TTE pretraining contributes to MOTOR's performance, we perform a comparison with the common pretraining task of next code prediction. Here pretraining predicts the next code in a patient's timeline, similar to autoregressive pretraining in NLP (see Appendix K for more details). We perform this ablation on the smaller STARR-OMOP dataset where pretraining is the most impactful.

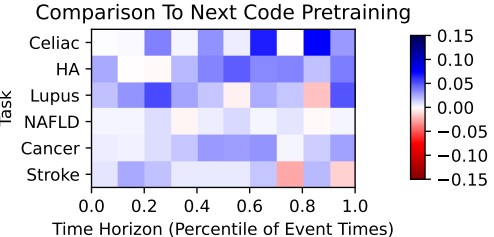

Table 2 compares the next code pretraining task to our TTE pretraining task on STARR-OMOP. The TTE pretraining task outperforms the next code pretraining task, as indicated by the statistically significant higher C statistics on four of the target tasks (confidence intervals in Table S6).

Figure 2: Per-time horizon performance differences (in time-dependent C statistic) when replacing autoregressive pretraining with TTE pretraining. Blue indicates better performance with TTE pretraining and red indicates worse.

In order to better understand how these objectives differ in performance, we also performed a time-horizon oriented evaluation where we evaluate the time-dependent C statistic within time horizon deciles, with results in Figure 2 (the same time horizon analysis for all baselines is in Appendix J.5). We observe benefits of TTE pretraining for most time horizons and tasks.

Table 2: Impact of the pretraining objective on model performance in terms of the time-dependent C statistic.

| Objective | Celiac | HA | Lupus | NAFLD | Cancer | Stroke |
|---|---|---|---|---|---|---|
| Next Code | 0.774 | 0.862 | 0.842 | 0.860 | 0.860 | 0.857 |
| Time-to-Event | **0.802** | **0.887** | **0.863** | **0.864** | **0.865** | **0.875** |

**Hyperparameter Ablations:** MOTOR has a variety of important hyperparameters that all potentially contribute to performance. In order to better understand MOTOR, we have performed a series of hyperparameter ablations on the STARR-OMOP dataset using linear probe adaption on the code-based tasks. Figure 3 shows the performance when ablating: the number of pretraining tasks, the attention width, and the number of pieces in the piecewise exponential model. We find that MOTOR is robust to small changes in these hyperparameters but that performance degrades when all three are majorly reduced, showing the validity of our chosen hyperparameters.

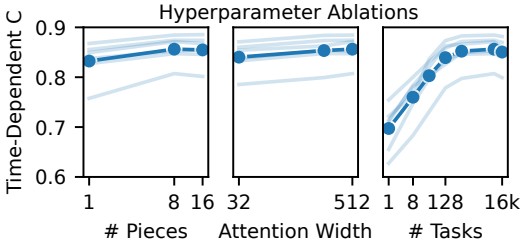

Figure 3: Performance for code-based tasks on STARR-OMOP when tuning the number of piecewise pieces, attention width, and number of pretraining tasks. Lines are average performance (blue) and per-task performance (light blue).

**Robustness Across Time And Portability Across Datasets:** Two key questions are whether MOTOR is robust to shifts over time and whether it is portable to other sites (e.g., whether a pretrained MOTOR model can be reused on data from other hospitals). We test robustness across time by performing an out-of-time evaluation using a prospective sample constructed from nine months of the most recent STARR-OMOP data. Likewise, to address model portability, we perform experiments on MIMIC-IV (Johnson et al., 2023) where we transfer a MOTOR model that was pretrained on STARR-OMOP to MIMIC-IV using linear probe adaptation (MOTOR-Transfer Probe). We compare that adapted model to baselines trained from scratch on MIMIC-IV. Table 3 contains the best baseline and MOTOR time-dependent C statistics for each of these experiments. We find that MOTOR outperforms all baselines. Further details and the full tables for these experiments can be found in Appendices J.6 and J.7.

Table 3: Time-dependent C statistic on an out-of-time STARR-OMOP sample and MIMIC-IV.

| Method | Dataset | Celiac | HA | Lupus | NAFLD | Cancer | Stroke |
|---|---|---|---|---|---|---|---|
| Best Baseline | Out-of-time STARR-OMOP | 0.682 | 0.776 | 0.810 | 0.750 | 0.786 | 0.768 |
| Best MOTOR | - | **0.792** | **0.843** | **0.880** | **0.809** | **0.870** | **0.836** |
| Best Baseline | MIMIC-IV | 0.625 | 0.820 | 0.807 | 0.736 | 0.748 | 0.780 |
| Best MOTOR | - | **0.628** | **0.850** | **0.819** | **0.802** | **0.828** | **0.812** |

**Sample Efficiency:** A key benefit of foundation models is improved sample efficiency, being able to learn task models with few labels. We evaluate MOTOR's sample efficiency in two ways: first, how performance changes as we pretrain with fewer patients and, second, how it changes as we adapt with fewer patients.

Figure 4 contains the results of these experiments using the linear probe adaptation strategy. We use 5%, 10%, and 25% samples of the dataset for both the pretraining and fine-tuning experiments. We find that MOTOR is

robust to the amount of adaptation data available, performing well with limited task labels. However, MOTOR performance degrades quite rapidly when the pretraining dataset is reduced. This aligns with the intuition on the importance of pretraining in foundation models. In the same theme of investigating low resource environments, we also evaluate the compute efficiency of adapting MOTOR in Appendix J.8 and find that MOTOR-Linear Probe is ten times faster than RSF, the strongest baseline.

## 6 DISCUSSION

We used TTE pretraining to create the MOTOR foundation model and have reported its performance for a wide variety of time-to-event tasks across datasets. The process of pretraining allows us to use a high-parameter (143M parameters) model even with limited data. Our results show that MOTOR-derived models outperform state-of-the-art approaches across a range of clinical tasks. This success is owed to the novel TTE pretraining objective that enables MOTOR to learn sophisticated interactions and event distributions.

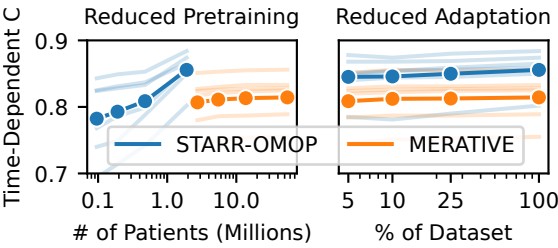

Figure 4: Performance when MOTOR is pretrained or adapted with less data. Lines are average performance (blue) and per-task performance (light blue).

We have also verified that MOTOR has many of the desirable foundation model properties. It is computational and label efficient in that it can be adapted to downstream TTE tasks with minimal compute and labeled data (only requiring 1/10th the time and 5% of the labeled task data to match RSF). It is robust over both time and can be reused in other datasets, with out-of-time and MIMIC-IV experiments showing better predictive performance than all baselines. Finally, MOTOR is general purpose, with demonstrated performance for 19 tasks, including text based tasks that are completely out of the domain of the pretraining tasks.

This work has limitations. First, pretraining MOTOR is expensive, requiring up to two GPU weeks for MERATIVE. Second, our experiments perform evaluations on subsampled target task datasets, which introduces censoring bias (as discussed in Appendix D), which might affect our results. Third, our handling of competing risks (censoring at death) may not be appropriate for every setting.

Finally, to enable reproducibility, which is hampered (McDermott et al., 2021; TSIMA, 2023) by challenges in releasing datasets and large pretrained models (Wornow et al., 2023), we release a 143M parameter MOTOR model pretrained on STARR-OMOP under a research DUA, which prohibits re-identifying patients as well as commercial and non-research use. We discuss further details, including safeguards, in Appendix L.

## 7 CONCLUSION

Time-to-event prediction is an important but challenging problem in the medical setting. A foundation model pre-trained with the TTE objective can enable transfer learning across datasets. We have pretrained such a model - MOTOR - on 55M patient records spanning 9B events and demonstrated transfer learning ability for 19 tasks across 3 datasets. The use of pretraining, and in particular specialized TTE pretraining, allows us to learn a TTE foundation model that can be adapted even in low label settings. Task-specific models based on MOTOR improved time-dependent C statistic by 4.6% over state-of-the-art, were more robust to temporal distributional shifts, and outperformed all baselines when porting across datasets.

## ACKNOWLEDGEMENTS

This work was supported by R01 HL144555 from the National Heart, Lung, and Blood Institute (NHLBI). Jason Fries was supported in part by an AIMI-HAI Partnership Grant. We would also like to thank Michael Warnow, Rahul Thapa, Scotty Fleming and Alejandro Schuler for providing useful comments on this paper.

## ETHICS STATEMENT

We follow current best practices to ensure that our use of structured medical data is safe, fair, and ethical. To protect patient privacy, we follow the standard practice for healthcare machine learning reproducibility established by PhysioNet (Goldberger et al., 2000) and Medical AI Research Foundations (Azizi et al., 2023), an open-source repository of medical foundation models, which provide several mechanisms to protect patient privacy. We summarize specific ethical considerations below and, where appropriate, provide links to expanded discussion in the appendix.

**Data Deidentification**: All data used for model training and evaluation were stripped of protected health information. STARR-OMOP is de-identified at the institutional level using the Safe Harbor standard to satisfy HIPAA's Privacy Rule. MERATIVE was de-identified by IBM to meet the limited data set HIPAA standard. All MOTOR model artifacts underwent an additional automated and manual human review process to verify the absence of incidental PHI. We have received permission from our academic institution to release this model given the model protections and data use agreement (DUA) outlined in Appendix L.

**Training Data Storage & Security**: All training data was stored in a HIPAA-compliant compute environment (Appendix C) with access limited to accredited individuals. MOTOR will be hosted in a gated environment similar to PhysioNet, where access requires a credentialed account and a non-commercial research DUA.

**Patient Consent**: For STARR-OMOP, all included patients from our medical center signed a privacy notice informing them that their records may be used for research purposes. De-identified data is made available to researchers via a school-wide IRB protocol. MERATIVE is provided to us in de-identified form, with assurance that it meets the requirements for a limited data set, which does not require patient consent for use in research.

**Algorithmic Bias**: We acknowledge that healthcare machine learning models are vulnerable to algorithmic bias, which can be especially problematic if used to guide treatment. Unfortunately, bias mitigation in medical foundation models, which prioritize learning generalizable feature representations that are useful for a wide range of tasks, is still an open research question (Jones et al., 2023) and not addressed by this work. Therefore, we take two steps to mitigate risk. First, as described in Appendix L, our model release comes with a DUA that expressly prohibits using MOTOR to guide medical care. Second, per the recommendations of (Chang et al., 2022), we perform an analysis in Appendix J.12 to characterize performance in sensitive subgroups to make sure that our model doesn't unfairly penalize sensitive groups compared to existing methods. We measure performance using the time-dependent C statistic on STARR-OMOP and compare MOTOR-Finetune and the strongest baseline RSF. We find that with one statistically insignificant exception, MOTOR-Finetune does not reduce performance within sensitive groups and instead broadly improves risk ranking across groups.

**Risk of Harm From Misuse**: Possible model misuse, e.g., for surveillance purposes, is a serious concern in domains like medicine. To mitigate these risks, we implement the following guardrails. (1) We restrict access to MOTOR weights via user credentialing and gated hosting akin to PhysioNet. (2) We prohibit medical decision making, diagnoses, and other non-research use cases of MOTOR in our DUA and can revoke model access for violations. (3) We do not release our MERATIVE models pretrained on insurance claims data, which may be easier for entities to obtain commercially vs. longitudinal EHR data.

## REPRODUCIBILITY STATEMENT

The code necessary for reproducing the experiments in this paper can be found in the supplemental materials as a zip file. Our hyperparameter search grids can be found in Appendix A. To aid reproducibility we have performed evaluations on the public MIMIC-IV dataset (see Appendix J.7) using a pretrained MOTOR model that we will be releasing (see Appendix L). It is important to note that while we do have approval for model release, our approved protocol does not allow release to anonymous reviewers. In addition, due to our data usage agreements, we are not allowed to share the source training datasets for STARR-OMOP or MERATIVE, however MERATIVE is commercially available.

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

# APPENDICES

## APPENDIX A    HYPERPARAMETER SEARCH GRIDS

Table S1: Hyperparmeter search grids and software versions of methods under comparison.

| Hyperparameters | Values |
|---|---|
| **Cox PH** | |
| lambda | automatic |
| min_patients_per_feature | 100, 1000 |
| software_version | glmnet 4.1-6 |
| **DeepSurv** | |
| dropout | 0, 0.1 |
| first_layer_size | 32, 128, 256, 512 |
| second_layer_size | 32 |
| learning_rate | $10^{-3}, 10^{-4}, 10^{-5}$ |
| min_patients_per_feature | 100, 1000 |
| software_version | pycox 0.2.3 |
| **DeepSurvivalMachines** | |
| first_layer_size | 32, 128, 256, 512 |
| second_layer_size | 32 |
| learning_rate | $10^{-3}, 10^{-4}, 10^{-5}$ |
| k | 3, 4, 6 |
| distribution | LogNormal, Weibull |
| min_patients_per_feature | 100, 1000 |
| software_version | auton-survival 0.0.0 |
| **DeepHit** | |
| dropout | 0, 0.1 |
| first_layer_size | 32, 128, 256, 512 |
| second_layer_size | 32 |
| learning_rate | $10^{-3}, 10^{-4}, 10^{-5}$ |
| num_bins | 8, 16, 32 |
| min_patients_per_feature | 100, 1000 |
| software_version | pycox 0.2.3 |
| **Random Survival Forests** | |
| node_size | 5 , 15, 50, 100 |
| min_time_bins | 8, 16, 32 |
| min_patients_per_feature | 100, 1000 |
| software_version | randomForestSRC 3.1.0 |
| **MOTOR** | |
| dropout | 0, 0.1 |
| learning_rate | $10^{-3}, 10^{-4}, 10^{-5}$ |
| num_time_pieces | 8, 16 |
| survival_dim | 256, 512 |
| inner_dim | 768 |
| layers | 6, 12 |
| max_sequence_length | 16,384 |
| vocabulary_size | 65,536 |

## APPENDIX B    DATA PROCESSING DETAILS

We first harmonize MERATIVE by converting it to the Observational Medical Outcomes Partnership (OMOP) schema (Overhage et al., 2012) (STARR-OMOP comes in the OMOP schema). We then process our datasets by converting all of the tables within the OMOP schema to a uniform timeline format of timestamped events, where each event is annotated with both the time and the clinical code for that event. We additionally perform some post-processing to correct several timestamp-related issues in the source data. In particular, we move billing codes to the end of visits to better reflect when those billing codes are clinically available and move all events annotated at 12:00 AM to 11:59 PM to better account for how 12:00 AM timestamps frequently represent day-level timestamps and we do not want to have that data available to models before the end of the day. We also adjust for events before birth by moving them to the date of birth.

To obtain unbiased evaluation metrics for the performance of methods, we randomly split our data into training, validation, and test sets, containing 70%, 15% and 15% of the patients in the dataset. We use a hash-based splitting algorithm so that splits are stable across dataset versions. The training set is used for all model training, the validation set is used for hyperparameter selection, and the test set is used for out-of-sample evaluation.

Table S2 contains the number of patients and events in each split for both datasets.

Table S2: Dataset split details

| Split Name | STARR-OMOP | | MERATIVE | |
|---|---|---|---|---|
| | # Patients | # Events | # Patients | # Events |
| Training Set | 1,910,887 | 3,149,948,337 | 54,721,282 | 8,976,493,594 |
| Validation Set | 409,704 | 672,623,024 | 11,719,566 | 1,922,762,263 |
| Test Set | 409,820 | 673,687,781 | 11,726,429 | 1,923,815,575 |

We also compute some summary statistics for both of our datasets, which are shown in Table S3.

Table S3: Demographic statistics for STARR-OMOP and MERATIVE. All patients have unknown race in MERATIVE as race is not provided. Standard deviations are provided in parenthesis when applicable.

| Statistic Name | STARR-OMOP | MERATIVE |
|---|---|---|
| Total Patients | 2,730,411 | 78,167,277 |
| Total Events | 4,496,259,142 | 12,823,071,432 |
| Average Events Per Patient | 1,646.7 (7,356.3) | 164.0 (241.8) |
| Average Age | 42.4 (24.3) | 35.9 (19.0) |
| % Male | 45.1 | 45.0 |
| % American Indian or Alaska Native | 0.3 | N/A |
| % Asian | 15.7 | N/A |
| % African American | 3.5 | N/A |
| % Native Hawaiian or Other Pacific Islander | 1.0 | N/A |
| % White | 42.9 | N/A |
| % Unknown | 36.7 | 100 |

## APPENDIX C    COMPUTE ENVIRONMENT

Experiments are performed in a local on-prem university compute environment using 24 Intel Xeon 2.70GHz CPU cores and 8 Nvidia V100 GPUs. All compute environments supported HIPAA-compliant data protocols., with access limited to accredited individuals. Each pretraining and/or finetuning job is only assigned one V100 at a time. Pretraining on STARR-OMOP takes 111 V100 GPU hours, while pretraining on MERATIVE takes 330 V100 GPU hours.

## APPENDIX D    TARGET TASK SUBSAMPLING

Our task definition, of predicting the time until a coded event given a year of history, would generate a single label per patient in our dataset. This produces millions of labels for STARR-OMOP (tens of millions for MERATIVE) and is not feasible to analyze with most existing TTE modeling approaches. In order to work around this, in our experiments, we use a simple subsampling strategy where we drop a fraction $d$ of the censored examples in our dataset before modeling and evaluation. This causes a non-uniform scaling of the hazard rates. However, this does not affect our primary results as the resulting hazard rate scaling bias is shared across all methods and the primary goal of this study is to evaluate the relative performance of different modeling approaches. Since we evaluate and train all models using the same subsampling, the censoring bias affects them in a similar manner and our ranking of models' performance should not be impacted. We considered correcting for this bias with a weighting strategy (Kohler et al., 2002; Van der Laan & Robins, 2003), but extreme weights induce unstable training. We additionally do one other step of subsampling, where if the number of labels is still over 200,000 despite the prior strategy, we simply select a random sample of 200,000 when training models.

Even though subsampling controls in principle shouldn't affect our ranking of methods, it is still useful to derive the exact hazard rate scaling induced by it. Let $H_i(t)$ be the hazard rate for a particular time t and let $\widetilde{H_i}(t)$ be the hazard rate for a particular time $t$ induced by our subsampling strategy. Equation 2 provides the ratio of $\widetilde{H_i}(t)$ and $H_i(t)$. Proof D is the derivation of that equation.

$$\frac{\widetilde{H_i}(t)}{H_i(t)} = \frac{1}{1 - d\,\mathbb{P}(C_i < T_i | C_i > t, T_i > t)} \tag{2}$$

*Proof.* The hazard in our original dataset before sampling is

$$
\begin{aligned}
H_i(t) &= \frac{\mathbb{P}(T_i = t)}{\mathbb{P}(T_i > t)} \\
&= \frac{\mathbb{P}(T_i = t)\,\mathbb{P}(C_i > t)}{\mathbb{P}(T_i > t)\,\mathbb{P}(C_i > t)} \\
&= \frac{\mathbb{P}(T_i = t, C_i > t)}{\mathbb{P}(T_i > t, C_i > t)} \qquad \text{assume } C_i \perp\!\!\!\perp T_i \\
&= \frac{\mathbb{P}(T_i = t, C_i > t)}{\mathbb{P}(C_i < T_i, C_i > t, T_i > t) + \mathbb{P}(C_i \geq T_i, C_i > t, T_i > t)}
\end{aligned}
$$

The hazard in our dataset after sampling is

$$
\begin{aligned}
\widetilde{H}_i(t) &= \frac{\mathbb{P}(T_i = t, C_i > t)}{(1-d)\,\mathbb{P}(C_i < T_i, T_i > t, C_i > t) + \mathbb{P}(C_i \geq T_i, T_i > t, C_i > t)} \\
&= \frac{\mathbb{P}(T_i = t, C_i > t)}{((1-d)\,\mathbb{P}(C_i < T_i | T_i > t, C_i > t) + \mathbb{P}(C_i \geq T_i | T_i > t, C_i > t))\,\mathbb{P}(T_i > t, C_i > t)} \\
&= \frac{\mathbb{P}(T_i = t, C_i > t)}{((1-d)\,\mathbb{P}(C_i < T_i | T_i > t, C_i > t) + (1 - \mathbb{P}(C_i < T_i | T_i > t, C_i > t)))\,\mathbb{P}(T_i > t, C_i > t)} \\
&= \frac{\mathbb{P}(T_i = t, C_i > t)}{(1 - d\,\mathbb{P}(C_i < T_i | T_i > t, C_i > t))\,\mathbb{P}(T_i > t, C_i > t)} \\
&= \frac{H_i(t)}{1 - d\,\mathbb{P}(C_i < T_i | T_i > t, C_i > t)}
\end{aligned}
$$

Thus,

$$
\frac{\widetilde{H}_i(t)}{H_i(t)} = \frac{1}{1 - d\,\mathbb{P}(C_i < T_i | C_i > t, T_i > t)}
$$

$\square$

## APPENDIX E  TARGET TASK DETAILS

Table S4: Task details for six code based tasks on STARR-OMOP and MERATIVE. We report the number of cases, the number of controls, the median event time, and the 90th percentile event time for each task. Times are reported in years.

| Task Name | ICD 10 Codes | STARR-OMOP | | MERATIVE | |
|---|---|---|---|---|---|
| | | # Cases | # Controls | # Cases | # Controls |
| Celiac Disease | K90.0 | 3,453 | 13,812 | 91,119 | 364,476 |
| Heart Attack | I21.* | 2,815 | 11,260 | 116,085 | 464,340 |
| Lupus | M32.* | 3,274 | 13,812 | 90,002 | 360,008 |
| Pancreatic Cancer | C25.* | 2,163 | 8,652 | 16,175 | 64,700 |
| NAFLD | K76.0 | 22,013 | 88,052 | 666,345 | 2,665,380 |
| Stroke | I63.* | 11,855 | 47,420 | 43,338 | 173,352 |

| Task Name | STARR-OMOP | | MERATIVE | |
|---|---|---|---|---|
| | Median Time | 90th Percentile Time | Median Time | 90th Percentile Time |
| Celiac Disease | 0.540 | 4.254 | 0.496 | 2.458 |
| Heart Attack | 0.512 | 4.824 | 0.545 | 2.606 |
| Lupus | 0.572 | 5.202 | 0.455 | 2.301 |
| Pancreatic Cancer | 0.464 | 5.511 | 0.323 | 2.329 |
| NAFLD | 0.862 | 5.293 | 0.496 | 2.521 |
| Stroke | 0.900 | 5.502 | 0.399 | 2.126 |

## APPENDIX F  TRANSFORMER DETAILS

Following the prior work of BEHRT (Li et al., 2020), we use a Transformer-based neural network for processing our medical record data. $X_i$, the sequence of codes and timestamps for patient $i$, is first transformed into a combination of code and time embeddings. Just as in BEHRT, missing data is not explicitly handled (since no codes are generated when data is missing), but time embeddings should in principle provide the necessary signal for learning important information about missingness. Code embeddings are generated with a standard embedding layer. Time embeddings are generated using a rotary position embedding (Su et al., 2024) with time (in days since birth) instead of the more standard position index. These embeddings are then fed into a Transformer to generate representations for every event in each patient's record.

Our Transformer differs from the main prior work of BEHRT in four major aspects. First, unlike BEHRT, which uses a masked language modeling objective that incorporates bidirectional information from patient sequences, we use causally masked attention (Dai & Le, 2015; Peters et al., 2018; Raffel et al., 2022). This attention pattern ensures information only flows forwards in the sequence and not backwards to guarantee that the model does not "cheat" by using information from the future. If fully connected attention is used instead, models quickly learn shortcuts to use future information to predict the past, which then results in failure when the models are applied to real data. Second, we use rotary position embeddings (Su et al., 2024) for combining the time and code embeddings at every attention subunit in the network. Third, we use a local attention mechanism, where attention is limited to 512 token context windows, to enable the use of sequence

lengths up to 16,384 events and avoid quadratic complexity (Roy et al., 2021). Fourth, we do not use the explicit visit-based components of BEHRT (the segment and position features) as our source medical record data contains overlapping visits that are not compatible with BEHRT's approach. Instead, we represent visits as additional clinical events within the timeline.

## APPENDIX G  BASELINE DETAILS

In order to verify the performance benefits of our method, we compared it to five commonly used state-of-art time-to-event approaches: Cox PH, DeepSurv, DeepHit, DSM, and RSF.

Cox PH and RSF cannot naively handle our sequential event format as they require input features to be in the form of a feature matrix. In order to run those methods on our dataset, we implement a featurization strategy to transform our timestamped event data into rows and columns that can be used for those methods. Following prior standard practice in the medical domain (OHDSI, 2019), we implement count featurization, where a column is created for each event code in the source data and each value in that column corresponds to the number of times that event code appears in the patient record.

In principle, DeepSurv, DeepHit, and DSM can accept the same event/timestamp input format as MOTOR by using a Transformer, however we observe poor performance due to overfitting on our small task datasets. Instead, we train these three baselines on the same feature matrices used for RSF and Cox PH. Training in this way requires learning many fewer parameters and thus is less prone to overfitting. We parameterize these three models with feedforward neural networks with two hidden layers, rectified linear unit (ReLU) activation functions, and batch-normalization. As in MOTOR, we use dropout and early stopping for regularization.

Appendix A contains the hyperparameter search grids and software versions for each baseline method.

## APPENDIX H  EVALUATION METRICS DETAILS

The time-dependent C statistic summarizes the ability of a predictive model to rank patients by risk based on their instantaneous hazard functions. Specifically, we apply the incident and dynamic definitions for the time-dependent sensitivity and specificity, respectively, following the choices in Heagerty & Zheng (2005):

$$\text{sen}^{\mathbb{I}}(c, t) = \mathbb{P}(S_i > c | \mathrm{d}N_i^*(t) = 1)$$

$$\text{spe}^{\mathbb{D}}(c, t) = \mathbb{P}(S_i \leq c | N_i^*(t) = 0),$$

where $S_i$ is a predicted risk score for subject $i$. $N_i^*(t) = \mathbb{1}\{T_i \leq t\}$ and $\mathrm{d}N_i^*(t) = N_i^*(t) - N_i^*(t-)$ follow the standard definitions of a counting process. Then, the concordance can be expressed as a weighted average of the area under the time-dependent receiver operating curve (ROC)

$$C_{td} = \frac{\int_t \text{AUC}(t) \, w(t) \, \mathrm{d}t}{\int_t w(t) \, \mathrm{d}t} \tag{3}$$

where $w(t) = f(t) \cdot S(t)$. $f(t)$ is the event rate at a particular time t and $S(t)$ is the survival probability at that time. Both $f(t)$ and $S(t)$ are estimated using the Kaplan-Meier estimator. $\text{AUC}(t)$ is the time-specific area under ROC and can be computed by integration, and ROC can be calculated using $\text{sen}^{\mathbb{I}}$ and $\text{spe}^{\mathbb{D}}$ with the standard formulation.

Note that the formula in Equation (3) is slightly modified from Heagerty & Zheng (2005) because our dataset contains many events that happened at the exact same time due to some events having only day-level time resolution. Heagerty & Zheng (2005) assumes that all event times are distinct thus the sum of the weights (the term in the denominator) is $\frac{1}{2}$. As we have many tied event times, we cannot make the same assumption and have to explicitly compute the denominator as $\int_t w(t) \, \mathrm{d}t$.

Harrell's C-index (Harrell et al., 1982) is an alternative type of C statistic that summarizes the ability of a predictive model to rank patients. There are two primary limitations of the Harrell's C-index that the time-dependent C statistic does not have (Hartman et al., 2023). First, Harrell's C-index does not account for censoring so it will be biased when used for evaluating TTE modeling performance. Second, Harrell's C-index requires the predictive model to output a single risk per patient, as opposed to risk over time. This causes problems when evaluating non-proportional hazards models such as RSF and MOTOR that have time-varying relative risks. We adjust for this by using the average hazard for models with time-varying risks. This will underestimate the performance of RSF and MOTOR, but will allow us to apply this metric.

We then use the standard formulation for computing Harrell's C-index. First, we find all possible pairs $P$, where one patient is known to have the event before another, and split that set into $P_{\text{correct}}$ (if the higher risk patient has the event first), $P_{\text{tied}}$ (if both patients have the same risk), and $P_{\text{incorrect}}$ (if the lower risk patient has the event first).

$$C_H = \frac{P_{\text{correct}} + 0.5 \times P_{\text{tied}}}{P_{\text{correct}} + P_{\text{tied}} + P_{\text{incorrect}}} \tag{4}$$

The Nam-D'Agostino (ND) calibration metric (D'Agostino & Nam, 2003) is constructed based on the idea that the survival probability estimates, $\hat{S}(t) = P(T_i > t)$, for a particular time $t$ should be calibrated with respect to the number of observed survivors past that time point, which is not available in censored data, thus we estimate it using the Kaplan-Meier estimator per D'Agostino & Nam (2003). First, the data is sorted by $\hat{S}(t)$ given by a model and split into $M$ equal size risk bins, where $\overline{p(t)}_m$ is the average value of $\hat{S}(t)$ within bin $m$. Second, a KM estimator is fit within each risk bin for computing actual survival probabilities. Finally, the squared difference between the expected and actual values is computed and summed across bins. The resulting statistic is $\chi^2$ with $M-1$ degrees of freedom. In principle, you can run hypothesis tests using this statistic, but we instead directly report and compare the following raw statistic between models:

$$\chi^2(t) = \sum_{m=1}^{M} \frac{[KM(t)_m - \overline{p(t)}_m]^2 n_m}{\overline{p(t)}_m(1 - \overline{p(t)}_m)}, \tag{5}$$

where $n_m$ is the number of observations in bin $m$ and $n_1 = n_2 = ... = n_M$. For all evaluations, we set $t$ to the median event time. In addition, we make one minor modification to this metric in that we remove the $n_m$ term in the numerator as we are primarily concerned about relative performance. $n_m$ a constant factor that only depends on the number of patients for each task, and removing it makes it easier to compare results across target tasks.

The Integrated Brier Score (IBS) (Graf et al., 1999) simultaneously measures the ability of a time-to-event model to both rank patients correctly and have calibrated estimates. We directly use the implementation of IBS provided by the scikit-survival package (Pölsterl, 2020), which integrates the standard Brier Score metric using numeric integration. For all of our experiments, we integrate from the 10th to the 90th percentile of event times, using 256 trapezoids to estimate the integral.

The time-dependent C statistic and ND test become increasingly unstable as the censoring rate increases since they rely on Kaplan Meier estimators that themselves become unstable at higher censoring rates. As such, it is generally recommended to only compute these statistics on a subset of the possible times to reduce the variance of estimates. As suggested by Heagerty & Zheng (2005), we use the time-restricted variants of these evaluation metrics by setting a time limit of the 90th percentile of observed event times and only perform evaluations up to that time point. We report the 90th percentile time for each task on each dataset in Appendix E.

In order to accurately quantify statistical significance, we use the paired test-set bootstrap approach (Konietschke & Pauly, 2013) to estimate the 95% confidence interval for the difference in model performance

between MOTOR and every other method. For every target task, we resample our test set with replacement to obtain 1000 bootstrap samples. We then compute all metrics for all of our models on each bootstrap sample, subtracting MOTOR's performance each time to obtain an estimate of how each method differs from MOTOR. We finally compute confidence intervals by extracting the 2.5% and 97.5% percentiles of the bootstrapped samples. All of our confidence intervals for the differences in performance between methods are reported in Appendix I. As an additional aid, we also report standard deviations using the same bootstraping approach in Appendex M.

## APPENDIX I   CONFIDENCE INTERVALS

Table S5: 95% confidence intervals of difference in time-dependent C statistic between MOTOR-Finetune and the other methods. Higher numbers are considered better for this metric. ∗ indicates statistically significant entries at p = 0.05.

| Method | Celiac | Heart Attack | Lupus | NAFLD | Pancretic Cancer | Stroke |
|---|---|---|---|---|---|---|
| | | | STARR-OMOP | | | |
| Cox PH | [-0.140, -0.088]* | [-0.146, -0.105]* | [-0.113, -0.073]* | [-0.147, -0.129]* | [-0.093, -0.050]* | [-0.105, -0.086]* |
| DeepSurv | [-0.120, -0.076]* | [-0.079, -0.048]* | [-0.091, -0.056]* | [-0.070, -0.057]* | [-0.075, -0.035]* | [-0.052, -0.037]* |
| DSM | [-0.115, -0.075]* | [-0.076, -0.043]* | [-0.097, -0.060]* | [-0.065, -0.054]* | [-0.078, -0.035]* | [-0.047, -0.034]* |
| DeepHit | [-0.132, -0.084]* | [-0.078, -0.043]* | [-0.074, -0.038]* | [-0.065, -0.052]* | [-0.079, -0.037]* | [-0.049, -0.035]* |
| RSF | [-0.096, -0.051]* | [-0.067, -0.036]* | [-0.094, -0.057]* | [-0.068, -0.055]* | [-0.060, -0.023]* | [-0.041, -0.027]* |
| MOTOR-Scratch | [-0.131, -0.083]* | [-0.114, -0.073]* | [-0.081, -0.041]* | [-0.047, -0.037]* | [-0.114, -0.064]* | [-0.050, -0.036]* |
| MOTOR-Probe | [-0.010, 0.009] | [-0.008, 0.001] | [-0.020, -0.005]* | [-0.008, -0.002]* | [-0.006, 0.004] | [-0.004, 0.003] |
| MOTOR-Finetune | [0.000, 0.000] | [0.000, 0.000] | [0.000, 0.000] | [0.000, 0.000] | [0.000, 0.000] | [0.000, 0.000] |
| | | | MERATIVE | | | |
| Cox PH | [-0.273, -0.251]* | [-0.051, -0.043]* | [-0.095, -0.085]* | [-0.069, -0.056]* | [-0.219, -0.203]* | [-0.107, -0.096]* |
| DeepSurv | [-0.048, -0.037]* | [-0.019, -0.013]* | [-0.034, -0.028]* | [-0.039, -0.028]* | [-0.036, -0.029]* | [-0.046, -0.038]* |
| DSM | [-0.041, -0.031]* | [-0.020, -0.015]* | [-0.029, -0.023]* | [-0.034, -0.024]* | [-0.033, -0.027]* | [-0.040, -0.032]* |
| DeepHit | [-0.045, -0.034]* | [-0.018, -0.014]* | [-0.032, -0.026]* | [-0.039, -0.028]* | [-0.036, -0.030]* | [-0.045, -0.036]* |
| RSF | [-0.062, -0.052]* | [-0.022, -0.017]* | [-0.037, -0.031]* | [-0.028, -0.019]* | [-0.039, -0.032]* | [-0.053, -0.045]* |
| MOTOR-Scratch | [-0.029, -0.021]* | [-0.011, -0.006]* | [-0.015, -0.011]* | [-0.015, -0.008]* | [-0.015, -0.010]* | [-0.023, -0.017]* |
| MOTOR-Probe | [-0.010, -0.004]* | [-0.003, 0.000] | [-0.008, -0.004]* | [-0.008, -0.002]* | [-0.011, -0.007]* | [-0.008, -0.004]* |
| MOTOR-Finetune | [0.000, 0.000] | [0.000, 0.000] | [0.000, 0.000] | [0.000, 0.000] | [0.000, 0.000] | [0.000, 0.000] |

Table S6: 95% confidence intervals for the difference in MOTOR's time-dependent C statistic on the code-based tasks when the pretraining objective is changed. We evaluate both a "next code" baseline as well as our proposed time-to-event objective. ∗ indicates statistically significant entries at p = 0.05.

| Objective | Celiac | Heart Attack | Lupus | NAFLD | Pancretic Cancer | Stroke |
|---|---|---|---|---|---|---|
| Next Code | [-0.045, -0.010]* | [-0.036, -0.013]* | [-0.034, -0.009]* | [-0.008, 0.000] | [-0.021, 0.010] | [-0.023, -0.013]* |
| Time-to-Event | [0.000, 0.000] | [0.000, 0.000] | [0.000, 0.000] | [0.000, 0.000] | [0.000, 0.000] | [0.000, 0.000] |

Table S7: 95% confidence intervals for the difference in ND calibration between MOTOR-Finetune and other methods. Lower numbers are better for this metric. ∗ indicates statistically significant entries at p = 0.05.

| Method | Celiac | Heart Attack | Lupus | NAFLD | Pancretic Cancer | Stroke |
|---|---|---|---|---|---|---|
| | | | STARR-OMOP | | | |
| Cox PH | [-0.123, 0.202] | [-0.005, 0.309] | [0.082, 0.473]* | [0.029, 0.114]* | [-0.174, 0.190] | [-0.062, 0.026] |
| DeepSurv | [-0.220, 0.058] | [-0.111, 0.165] | [-0.138, 0.115] | [2.564, 3.231]* | [-0.190, 0.171] | [-0.058, 0.019] |
| DSM | [0.014, 1.875]* | [0.126, 3.849]* | [-0.107, 0.484] | [-0.066, -0.003]* | [0.193, 8.457]* | [-0.024, 0.523] |
| DeepHit | [0.850, 4.997]* | [0.025, 0.807]* | [-0.030, 0.604] | [-0.027, 0.082] | [0.049, 1.597]* | [0.031, 0.377]* |
| RSF | [-0.021, 0.390] | [-0.029, 0.329] | [-0.179, 0.116] | [-0.004, 0.088] | [0.064, 0.481]* | [0.005, 0.121]* |
| MOTOR-Scratch | [2.540, 10.180]* | [2.110, 10.715]* | [10.933, 51.137]* | [-0.013, 0.070] | [2.839, 19.885]* | [-0.020, 0.134] |
| MOTOR-Probe | [-0.202, 0.043] | [-0.147, 0.063] | [-0.179, 0.037] | [-0.059, -0.004]* | [-0.134, 0.047] | [-0.061, 0.021] |
| MOTOR-Finetune | [0.000, 0.000] | [0.000, 0.000] | [0.000, 0.000] | [0.000, 0.000] | [0.000, 0.000] | [0.000, 0.000] |
| | | | MERATIVE | | | |
| Cox PH | [-0.147, -0.077]* | [0.036, 0.076]* | [0.070, 0.138]* | [0.130, 0.244]* | [0.350, 0.523]* | [0.059, 0.096]* |
| DeepSurv | [-0.093, -0.035]* | [-0.016, 0.010] | [18.721, 21.664]* | [0.001, 0.062]* | [-0.110, -0.046]* | [2.870, 3.455]* |
| DSM | [-0.082, -0.023]* | [0.006, 0.130]* | [0.061, 0.162]* | [0.033, 0.196]* | [-0.009, 0.087] | [0.009, 0.123]* |
| DeepHit | [-0.065, 0.037] | [0.063, 0.109]* | [0.045, 0.096]* | [0.042, 0.313]* | [-0.057, 0.017] | [0.028, 0.084]* |
| RSF | [-0.076, -0.017]* | [0.014, 0.060]* | [-0.004, 0.047] | [0.281, 0.436]* | [-0.037, 0.036] | [-0.004, 0.043] |
| MOTOR-Scratch | [-0.139, -0.061]* | [0.004, 0.057]* | [-0.021, 0.014] | [0.019, 0.091]* | [-0.026, 0.033] | [0.026, 0.061]* |
| MOTOR-Probe | [-0.131, -0.064]* | [-0.014, 0.007] | [-0.021, 0.019] | [-0.022, 0.025] | [-0.081, -0.010]* | [-0.016, 0.006] |
| MOTOR-Finetune | [0.000, 0.000] | [0.000, 0.000] | [0.000, 0.000] | [0.000, 0.000] | [0.000, 0.000] | [0.000, 0.000] |

Table S8: 95% confidence intervals of difference in Harrell C-index between MOTOR-Finetune and the other methods. Higher numbers are considered better for this metric. ∗ indicates statistically significant entries at p = 0.05.

| Method | Celiac | Heart Attack | Lupus | NAFLD | Pancretic Cancer | Stroke |
|---|---|---|---|---|---|---|
| | | | STARR-OMOP | | | |
| Cox PH | [-0.122, -0.069]* | [-0.162, -0.117]* | [-0.107, -0.068]* | [-0.148, -0.129]* | [-0.120, -0.074]* | [-0.111, -0.088]* |
| DeepSurv | [-0.100, -0.056]* | [-0.059, -0.035]* | [-0.077, -0.050]* | [-0.061, -0.047]* | [-0.074, -0.041]* | [-0.045, -0.029]* |
| DSM | [-0.091, -0.051]* | [-0.411, -0.384]* | [-0.089, -0.056]* | [-0.058, -0.046]* | [-0.080, -0.046]* | [-0.045, -0.031]* |
| DeepHit | [-0.130, -0.084]* | [-0.136, -0.092]* | [-0.145, -0.104]* | [-0.195, -0.176]* | [-0.246, -0.177]* | [-0.239, -0.210]* |
| RSF | [-0.072, -0.033]* | [-0.056, -0.032]* | [-0.079, -0.045]* | [-0.079, -0.065]* | [-0.063, -0.024]* | [-0.053, -0.036]* |
| MOTOR-Scratch | [-0.089, -0.049]* | [-0.078, -0.046]* | [-0.077, -0.042]* | [-0.045, -0.036]* | [-0.087, -0.044]* | [-0.047, -0.034]* |
| MOTOR-Probe | [-0.017, 0.001] | [-0.011, -0.002]* | [-0.029, -0.016]* | [-0.012, -0.006]* | [-0.004, 0.001] | [-0.006, 0.000] |
| MOTOR-Finetune | [0.000, 0.000] | [0.000, 0.000] | [0.000, 0.000] | [0.000, 0.000] | [0.000, 0.000] | [0.000, 0.000] |
| | | | MERATIVE | | | |
| Cox PH | [-0.292, -0.279]* | [-0.064, -0.055]* | [-0.110, -0.098]* | [-0.090, -0.076]* | [-0.218, -0.205]* | [-0.125, -0.112]* |
| DeepSurv | [-0.053, -0.043]* | [-0.021, -0.016]* | [-0.036, -0.030]* | [-0.040, -0.031]* | [-0.038, -0.031]* | [-0.049, -0.040]* |
| DSM | [-0.292, -0.279]* | [-0.344, -0.334]* | [-0.355, -0.345]* | [-0.394, -0.381]* | [-0.351, -0.341]* | [-0.311, -0.300]* |
| DeepHit | [-0.072, -0.061]* | [-0.034, -0.027]* | [-0.078, -0.067]* | [-0.176, -0.153]* | [-0.059, -0.051]* | [-0.062, -0.052]* |
| RSF | [-0.071, -0.061]* | [-0.027, -0.022]* | [-0.044, -0.037]* | [-0.039, -0.030]* | [-0.046, -0.039]* | [-0.054, -0.045]* |
| MOTOR-Scratch | [-0.024, -0.017]* | [-0.011, -0.007]* | [-0.010, -0.006]* | [-0.013, -0.007]* | [-0.016, -0.011]* | [-0.017, -0.011]* |
| MOTOR-Probe | [-0.018, -0.012]* | [-0.008, -0.005]* | [-0.013, -0.009]* | [-0.008, -0.004]* | [-0.018, -0.014]* | [-0.013, -0.009]* |
| MOTOR-Finetune | [0.000, 0.000] | [0.000, 0.000] | [0.000, 0.000] | [0.000, 0.000] | [0.000, 0.000] | [0.000, 0.000] |

Table S9: 95% confidence intervals of difference in IBS between MOTOR-Finetune and the other methods. Lower numbers are considered better for this metric. ∗ indicates statistically significant entries at p = 0.05.

| Method | Celiac | Heart Attack | Lupus | NAFLD | Pancretic Cancer | Stroke |
|---|---|---|---|---|---|---|
| | | | STARR-OMOP | | | |
| Cox PH | [0.015, 0.029]* | [0.030, 0.048]* | [0.025, 0.042]* | [0.042, 0.048]* | [0.034, 0.051]* | [0.027, 0.036]* |
| DeepSurv | [0.011, 0.023]* | [0.013, 0.028]* | [0.012, 0.027]* | [0.025, 0.031]* | [0.023, 0.039]* | [0.010, 0.017]* |
| DSM | [0.013, 0.025]* | [0.015, 0.029]* | [0.012, 0.026]* | [0.017, 0.022]* | [0.030, 0.044]* | [0.010, 0.016]* |
| DeepHit | [0.016, 0.030]* | [0.011, 0.025]* | [0.007, 0.024]* | [0.012, 0.017]* | [0.027, 0.047]* | [0.006, 0.012]* |
| RSF | [0.012, 0.025]* | [0.011, 0.025]* | [0.009, 0.022]* | [0.022, 0.027]* | [0.017, 0.031]* | [0.011, 0.018]* |
| MOTOR-Scratch | [0.028, 0.044]* | [0.031, 0.054]* | [0.020, 0.038]* | [0.013, 0.018]* | [0.039, 0.066]* | [0.014, 0.021]* |
| MOTOR-Probe | [-0.004, 0.002] | [-0.003, 0.003] | [0.005, 0.012]* | [-0.001, 0.002] | [0.000, 0.004]* | [-0.007, -0.003]* |
| MOTOR-Finetune | [0.000, 0.000] | [0.000, 0.000] | [0.000, 0.000] | [0.000, 0.000] | [0.000, 0.000] | [0.000, 0.000] |
| | | | MERATIVE | | | |
| Cox PH | [-0.273, -0.251]* | [-0.051, -0.043]* | [-0.095, -0.085]* | [-0.069, -0.056]* | [-0.219, -0.203]* | [-0.107, -0.096]* |
| DeepSurv | [-0.048, -0.037]* | [-0.019, -0.013]* | [-0.034, -0.028]* | [-0.039, -0.028]* | [-0.036, -0.029]* | [-0.046, -0.038]* |
| DSM | [0.009, 0.011]* | [0.007, 0.009]* | [0.008, 0.010]* | [0.013, 0.017]* | [0.008, 0.010]* | [0.009, 0.012]* |
| DeepHit | [0.012, 0.015]* | [0.008, 0.010]* | [0.010, 0.013]* | [0.018, 0.023]* | [0.011, 0.014]* | [0.008, 0.011]* |
| RSF | [-0.062, -0.052]* | [-0.022, -0.017]* | [-0.037, -0.031]* | [-0.028, -0.019]* | [-0.039, -0.032]* | [-0.053, -0.045]* |
| MOTOR-Scratch | [-0.029, -0.021]* | [-0.011, -0.006]* | [-0.015, -0.011]* | [-0.015, -0.008]* | [-0.015, -0.010]* | [-0.023, -0.017]* |
| MOTOR-Probe | [-0.010, -0.004]* | [-0.003, 0.000] | [-0.008, -0.004]* | [-0.008, -0.002]* | [-0.011, -0.007]* | [-0.008, -0.004]* |
| MOTOR-Finetune | [0.000, 0.000] | [0.000, 0.000] | [0.000, 0.000] | [0.000, 0.000] | [0.000, 0.000] | [0.000, 0.000] |

## APPENDIX J   ADDITIONAL EXPERIMENTS

### J.1   CALIBRATION PERFORMANCE

Table S10 provides ND calibration results for both baselines and MOTOR on our six code-based tasks. MOTOR yields the best calibration for most of the tasks; however, the improvements are not statistically significant given the overlapping confidence intervals in Table S7. Of note is that the MOTOR-Scratch baseline has substantially worse calibration, which is a common failure mode for neural network models (Guo et al., 2017).

Table S10: ND calibration for the various methods on the full datasets on the code-based tasks. Smaller values indicate smaller calibration errors. Bolding indicates the best performing model.

| Method | Dataset | Celiac | HA | Lupus | NAFLD | Cancer | Stroke |
|---|---|---|---|---|---|---|---|
| Cox PH | STARR-OMOP | 0.167 | 0.214 | 0.373 | 0.121 | 0.141 | 0.031 |
| DeepSurv | - | **0.041** | 0.089 | 0.102 | 2.911 | 0.081 | **0.023** |
| DSM | - | 0.541 | 0.953 | 0.155 | **0.012** | 1.186 | 0.114 |
| DeepHit | - | 1.997 | 0.234 | 0.179 | 0.050 | 0.512 | 0.183 |
| RSF | - | 0.299 | 0.225 | 0.095 | 0.091 | 0.345 | 0.112 |
| MOTOR-Scratch | - | 4.970 | 3.990 | 23.882 | 0.077 | 7.104 | 0.083 |
| MOTOR-Probe | - | 0.045 | **0.029** | **0.059** | 0.019 | **0.046** | 0.032 |
| MOTOR-Finetune | - | 0.092 | 0.031 | 0.109 | 0.043 | 0.059 | 0.046 |
| Cox PH | MERATIVE | 0.029 | 0.068 | 0.122 | 0.205 | 0.503 | 0.093 |
| DSM | - | 0.060 | 0.040 | 0.111 | 0.097 | 0.107 | 0.056 |
| DeepHit | - | 0.098 | 0.097 | 0.085 | 0.117 | 0.062 | 0.066 |
| DeepSurv | - | 0.048 | 0.007 | 20.001 | 0.051 | **0.005** | 3.158 |
| RSF | - | 0.064 | 0.048 | 0.039 | 0.381 | 0.081 | 0.032 |
| MOTOR-Scratch | - | **0.013** | 0.034 | 0.015 | 0.067 | 0.086 | 0.058 |
| MOTOR-Probe | - | 0.014 | **0.007** | 0.017 | 0.017 | 0.037 | **0.010** |
| MOTOR-Finetune | - | 0.108 | 0.009 | **0.014** | **0.016** | 0.075 | 0.015 |

### J.2   HARRELL'S C-INDEX

Table S11 provides Harrell C-index results for both baselines and MOTOR on our six code-based tasks. MOTOR yield the best Harrell C-index on all of the tasks and the differences are statistically significant (see Table S8 for the confidence intervals).

### J.3   INTEGRATED BRIER SCORE

Table S12 provides Integrated Brier Score (IBS) metrics for both baselines and MOTOR on our six code-based tasks. MOTOR yield the best IBS on all of the tasks and the differences are statistically significant (see Table S9 for the confidence intervals).

### J.4   PERFORMANCE ON TEXT-BASED TASKS

One limitation of our primary code-based task set is that all of the tasks are defined with code-based labels, which might not generalize to a more diverse set of applications. In order to better evaluate performance on more diverse tasks, we define some additional tasks derived from clinical notes on STARR-OMOP

Table S11: Harrell's C-index for all methods on the code-based tasks. Bolding indicates the best performing model.

| Method | Dataset | Celiac | HA | Lupus | NAFLD | Cancer | Stroke |
|---|---|---|---|---|---|---|---|
| Cox PH | STARR-OMOP | 0.704 | 0.758 | 0.790 | 0.727 | 0.798 | 0.777 |
| DeepSurv | - | 0.722 | 0.851 | 0.814 | 0.811 | 0.838 | 0.839 |
| DSM | - | 0.728 | 0.500 | 0.806 | 0.813 | 0.832 | 0.838 |
| DeepHit | - | 0.692 | 0.785 | 0.754 | 0.679 | 0.682 | 0.651 |
| RSF | - | 0.747 | 0.853 | 0.816 | 0.793 | 0.852 | 0.831 |
| MOTOR-Scratch | - | 0.730 | 0.837 | 0.819 | 0.824 | 0.830 | 0.836 |
| MOTOR-Probe | - | 0.792 | 0.891 | 0.855 | 0.856 | 0.894 | 0.873 |
| MOTOR-Finetune | - | **0.799** | **0.898** | **0.878** | **0.865** | **0.895** | **0.876** |
| Cox PH | MERATIVE | 0.500 | 0.779 | 0.746 | 0.805 | 0.635 | 0.687 |
| DeepSurv | - | 0.738 | 0.820 | 0.817 | 0.852 | 0.811 | 0.761 |
| DSM | - | 0.500 | 0.500 | 0.500 | 0.500 | 0.500 | 0.500 |
| DeepHit | - | 0.719 | 0.809 | 0.777 | 0.724 | 0.791 | 0.748 |
| RSF | - | 0.720 | 0.814 | 0.809 | 0.853 | 0.804 | 0.755 |
| MOTOR-Scratch | - | 0.765 | 0.830 | 0.842 | 0.878 | 0.833 | 0.792 |
| MOTOR-Probe | - | 0.771 | 0.833 | 0.839 | 0.882 | 0.830 | 0.795 |
| MOTOR-Finetune | - | **0.786** | **0.839** | **0.850** | **0.888** | **0.846** | **0.805** |

Table S12: The IBS for the various methods on the full datasets on the code-based tasks. Smaller values are better for this metric. Bolding indicates the best performing model.

| Method | Dataset | Celiac | HA | Lupus | NAFLD | Cancer | Stroke |
|---|---|---|---|---|---|---|---|
| Cox PH | STARR-OMOP | 0.142 | 0.136 | 0.136 | 0.141 | 0.125 | 0.129 |
| DeepSurv | - | 0.137 | 0.118 | 0.122 | 0.124 | 0.114 | 0.110 |
| DSM | - | 0.139 | 0.119 | 0.122 | 0.116 | 0.119 | 0.110 |
| DeepHit | - | 0.143 | 0.116 | 0.118 | 0.111 | 0.119 | 0.106 |
| RSF | - | 0.139 | 0.115 | 0.118 | 0.121 | 0.107 | 0.112 |
| MOTOR-Scratch | - | 0.156 | 0.140 | 0.132 | 0.112 | 0.136 | 0.114 |
| MOTOR-Probe | - | **0.119** | 0.098 | 0.112 | 0.096 | 0.085 | **0.092** |
| MOTOR-Finetune | - | 0.120 | **0.097** | **0.103** | **0.096** | **0.083** | 0.097 |
| Cox PH | MERATIVE | 0.137 | 0.115 | 0.119 | 0.112 | 0.136 | 0.126 |
| DeepSurv | - | 0.120 | 0.107 | 0.126 | 0.094 | 0.107 | 0.144 |
| DSM | - | 0.119 | 0.108 | 0.101 | 0.092 | 0.107 | 0.117 |
| DeepHit | - | 0.122 | 0.110 | 0.104 | 0.098 | 0.110 | 0.116 |
| RSF | - | 0.123 | 0.109 | 0.103 | 0.093 | 0.109 | 0.119 |
| MOTOR-Scratch | - | 0.114 | 0.104 | 0.095 | 0.082 | 0.100 | 0.112 |
| MOTOR-Probe | - | 0.113 | 0.102 | 0.096 | 0.080 | 0.102 | 0.109 |
| MOTOR-Finetune | - | **0.109** | **0.101** | **0.093** | **0.077** | **0.098** | **0.107** |

(MERATIVE does not contain textual notes so cannot be used to create tasks). Irvin et al. (2019) created labeling functions to extract imaging observations from chest x-ray radiology notes, designed and tested to identify 13 different categories including fractures, lung lesions, and the presence of support devices. We construct time-to-event tasks for each possible observation by predicting the time until one of these

observations from the end of a random visit. For example, we construct an "Edema" task that predicts the time until a radiology note has a positive present mention of "Edema". The list of tasks and number of cases / controls for each task are detailed in Table S13. Note that these tasks are generally much larger than the code-based tasks.

Table S13: Task setup details for text based tasks on STARR-OMOP.

| Task Name | # Cases | # Controls |
|---|---|---|
| Atelectasis | 28,465 | 113,860 |
| Cardiomegaly | 15,528 | 62,112 |
| Consolidation | 15,678 | 62,712 |
| Edema | 38,619 | 154,476 |
| Enlarged Cardiomediastinum | 10,778 | 43,112 |
| Fracture | 51,011 | 204,044 |
| Lung Lesion | 77,848 | 311,392 |
| Lung Opacity | 75,688 | 302,752 |
| Pleural Effusion | 46,255 | 185,020 |
| Pleural Other | 6,362 | 25,448 |
| Pneumonia | 17,165 | 68,660 |
| Pneumothorax | 9,123 | 3,6492 |
| Support Devices | 91,498 | 365,992 |

Due to the vastly increased size of these datasets, we only, we only compare MOTOR-Probe, RSF and Cox PH. Table S14 provides the C statistics of each evaluated method on each task.

Table S14: Time-dependent C statistic for various methods on the text based tasks. Larger values indicate better ranking performance.

| Task | Cox PH | RSF | MOTOR-Probe |
|---|---|---|---|
| Atelectasis | 0.773 | 0.858 | 0.880 |
| Cardiomegaly | 0.778 | 0.873 | 0.899 |
| Consolidation | 0.787 | 0.857 | 0.895 |
| Edema | 0.724 | 0.804 | 0.841 |
| Enlarged Cardiomediastinum | 0.733 | 0.823 | 0.863 |
| Fracture | 0.680 | 0.753 | 0.777 |
| Lung Lesion | 0.741 | 0.804 | 0.824 |
| Lung Opacity | 0.739 | 0.825 | 0.858 |
| Pleural Effusion | 0.730 | 0.808 | 0.836 |
| Pleural Other | 0.754 | 0.840 | 0.856 |
| Pneumonia | 0.781 | 0.813 | 0.853 |
| Pneumothorax | 0.781 | 0.883 | 0.893 |
| Support Devices | 0.699 | 0.811 | 0.840 |

Results are very similar to the main code-based tasks as the relative ranking of methods stays the same. MOTOR is better than RSF, which in turn is better than Cox PH. This confirms that MOTOR continues to perform well, even on a diverse out of range task set that is based on radiology text labels.

## J.5 Performance As a Function of Time Horizon

One interesting question is how the difference in ranking performance changes as a function of the time horizon. Figure S1 plots a heatmap of the difference between MOTOR-Finetune and our six baselines across different time horizons on STARR-OMOP. We see that the largest benefit of MOTOR is for short time horizons, but there is a general improvement across all time horizons. The decreased improvement in the later time horizons probably reflects a lower maximum potential performance for longer predictions as longer time horizons are fundamentally harder to predict.

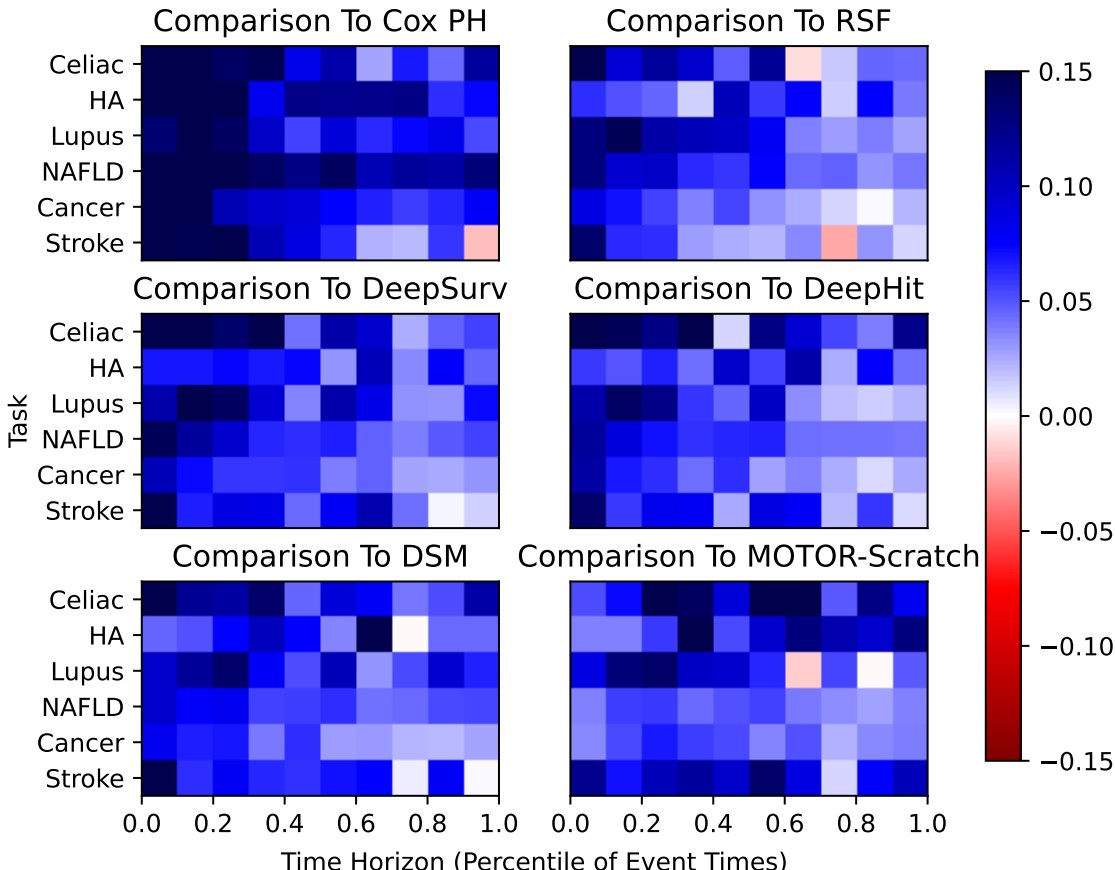

Figure S1: The difference in time-dependent C statistic between MOTOR-Finetune and various baselines on STARR-OMOP as a function of time-horizon.

## J.6 Out-of-time Performance

Our primary experiments consider out-of-patient evaluation, which measures how well models generalize to patients not used for training. However, it is often useful to also consider out-of-time evaluation, where models are evaluated on times outside the period where the models were trained. As noted in the main body

of our paper, all of our models were trained on STARR-OMOP data up until 2022/09/05. We have constructed a 9 month out-of-time evaluation set by obtaining additional STARR-OMOP data until 2023/06/11, labeling that data, then applying previously trained models to it. Table S15 contains the time-dependent C statistics for all models on this prospective set.

Table S15: Time-dependent C statistic for all methods on the code-based tasks on a prospective 9-month STARR-OMOP dataset. Bolding indicates the best performing model.

| Method | Celiac | HA | Lupus | NAFLD | Cancer | Stroke |
|---|---|---|---|---|---|---|
| Cox PH | 0.682 | 0.718 | 0.808 | 0.632 | 0.728 | 0.740 |
| DeepSurv | 0.663 | 0.703 | 0.799 | 0.689 | 0.767 | 0.767 |
| DSM | 0.657 | 0.776 | 0.810 | 0.715 | 0.786 | 0.768 |
| DeepHit | 0.585 | 0.553 | 0.589 | 0.540 | 0.653 | 0.558 |
| RSF | 0.651 | 0.719 | 0.793 | 0.654 | 0.744 | 0.728 |
| MOTOR-Scratch | 0.660 | 0.768 | 0.810 | 0.750 | 0.740 | 0.750 |
| MOTOR-Probe | 0.778 | 0.831 | 0.874 | 0.794 | 0.858 | **0.836** |
| MOTOR-Finetune | **0.792** | **0.843** | **0.880** | **0.809** | **0.870** | 0.833 |

We observe that all models appear to degrade in the prospective setting, but that MOTOR degrades to a lesser extent.

## J.7 MIMIC-IV TRANSFER

One important property of a foundation model is its ability to transfer across datasets. In order to evaluate cross-site transfer, we perform experiments where we transfer a MOTOR model that was pretrained on STARR-OMOP to MIMIC-IV 2.2 (Johnson et al., 2023) and perform linear probe adaptation on the smaller MIMIC-IV dataset. For comparison, we train baselines from scratch on MIMIC-IV. In order to process MIMIC-IV, we use the MIMIC-OMOP (MIMIC-OMOP) ETL to transform MIMIC-IV into OMOP and then process it with the same OMOP-based code used for the rest of our experiments. We evaluate using the same six code-based tasks, with the same task setup (i.e. predicting the time until the first diagnosis from the end of a random visit).

One major difference between MIMIC-IV and STARR-OMOP / MERATIVE is that MIMIC-IV is drastically smaller. Table S16 contains the label counts for the six code-based tasks on MIMIC-IV. To adjust for this in our baselines, we expand our hyperparameter grid with an additional entry allowing features that are only present in ten patients in addition to our existing one hundred and a thousand patient minimums.

Table S16: Task details for six code based tasks on MIMIC-IV.

| Task Name | ICD 10 Codes | # Cases | # Controls |
|---|---|---|---|
| Celiac Disease | K90.0 | 142 | 568 |
| Heart Attack | I21.* | 2,241 | 8,964 |
| Lupus | M32.* | 166 | 664 |
| Pancreatic Cancer | C25.* | 244 | 976 |
| NAFLD | K76.0 | 1,714 | 6,856 |
| Stroke | I63.* | 1,907 | 7,628 |

Table S17 contains the time-dependent C statistic for all baselines and MOTOR-Transfer Probe on MIMIC-IV, where MOTOR-Transfer Probe is the result of linear probe adaptation on a model that was pretrained on STARR-OMOP.

Table S17: Time-dependent C statistic for MOTOR-Transfer Probe vs baselines for the code-based tasks on MIMIC-IV. Bolding indicates the best performing model.

| Method | Celiac | HA | Lupus | NAFLD | Cancer | Stroke |
|---|---|---|---|---|---|---|
| Cox PH | 0.620 | 0.783 | 0.631 | 0.691 | 0.673 | 0.730 |
| DeepSurv | 0.625 | 0.805 | 0.807 | 0.702 | 0.748 | 0.738 |
| DSM | 0.561 | 0.820 | 0.679 | 0.692 | 0.598 | 0.754 |
| DeepHit | 0.618 | 0.808 | 0.754 | 0.670 | 0.646 | 0.750 |
| RSF | 0.527 | 0.817 | 0.717 | 0.736 | 0.656 | 0.780 |
| MOTOR-Transfer Probe | **0.628** | **0.850** | **0.819** | **0.802** | **0.828** | **0.812** |

### J.8 TIME TO TRAIN

The ability to fit TTE models quickly is essential for a variety of scenarios such as interactive model training and limited resource environments. One of the main advantages of MOTOR is that almost all of the weights can be frozen during finetuning, which means that one can learn target task models very quickly by only computing gradients for the linear heads.

In order to evaluate model training time, we perform timing experiments to compare the state of the art, RSF, to MOTOR in the fast linear probe setting. To simplify our experimental setup, we ignore the cost of hyperparameter tuning and focus solely on the time to fit a single final target task models with the optimally determined hyperparameters. In order to estimate the time to train as a function of sample size, we perform all experiments on random subsets of the NALFD task with the STARR-OMOP dataset, varying the training set size with each run. We provide 16 CPU cores, one V100 GPU, and 100 GB of RAM on isolated machines to both RSF and MOTOR.

Figure S2 plots the clock time for both methods as a function of the number of labels (with the farthest right point representing the size of the NALFD task). RSF is competitive in instances with fewer labels, where the overhead of MOTOR is slightly higher, but becomes much slower with increased label counts where the increased efficiency of MOTOR allows us to train target task models much more quickly. In the most extreme case, when we use the full NALFD task, MOTOR is 10x faster, achieving an average throughput of 147 patients per second (almost all of which is spent in representation generation). We note this is a somewhat conservative time comparison that ignores the cost and additional workflow complexity of materializing RSF's feature matrix.

### J.9 PERFORMANCE OF PRETRAINING TASK

When MOTOR is being pretrained, we are implicitly training 8,192 piecewise exponential TTE models across a variety of medical codes. Assessing pretraining performance is important because it allows us to evaluate MOTOR's performance on a wider variety of tasks than the limited subset we explicitly evaluated on. Figure S3 provides the time-dependent C statistic as a function of the amount of uncensored data for each task, with lines indicating the first quartile, median, and third quartile.

Performance on the pretraining tasks is strong across the vast majority of tasks, as indicated by the relatively high median C statistic of 0.842 for STARR-OMOP and 0.806 for MERATIVE. It is noteworthy that

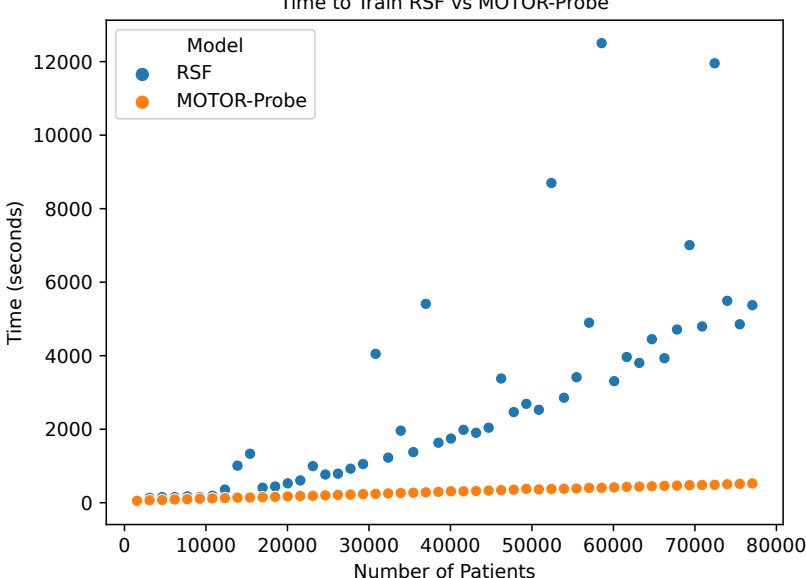

Figure S2: The time it takes to train target task models with MOTOR compared to RSF on varying amounts of patients.

performance is also strong for highly censored tasks, even when only one in a thousand patients have an uncensored event.

### J.10 CENSORING RATE WHEN FINETUNING

In order to better evaluate less efficient TTE methods, we artificially lower the censoring rate to 80% for all of our tasks. However, many tasks in real life have a censoring rate higher than 80%. In order to evaluate in this setting, we perform some experiments where we remove some of the uncensored examples in our dataset to increase the censoring rate. For the sake of time and compute, we only perform this analysis on STARR-OMOP.

Table S18 contains the time-dependent C statistics when MOTOR is fine-tuned using increasingly censored datasets. As the censoring rate increases, the performance decreases but only by an average of 1% even at 98% censored. This indicates that MOTOR performs fairly well even with highly censored data.

Table S18: Impact of the censoring rate when fine-tuning MOTOR on the time-dependent C statistic. We artificially increase the censoring rate by discarding increasing amounts of events. Experiments performed with STARR-OMOP.

| % Censored | Celiac | Heart Attack | Lupus | NAFLD | Pancretic Cancer | Stroke |
|---|---|---|---|---|---|---|
| 98% | 0.792 | 0.882 | 0.844 | 0.851 | 0.863 | 0.863 |
| 95% | 0.800 | 0.883 | 0.847 | 0.855 | 0.863 | 0.865 |
| 90% | 0.801 | 0.884 | 0.850 | 0.858 | 0.864 | 0.870 |
| 80% | **0.802** | **0.884** | **0.850** | **0.859** | **0.865** | **0.874** |

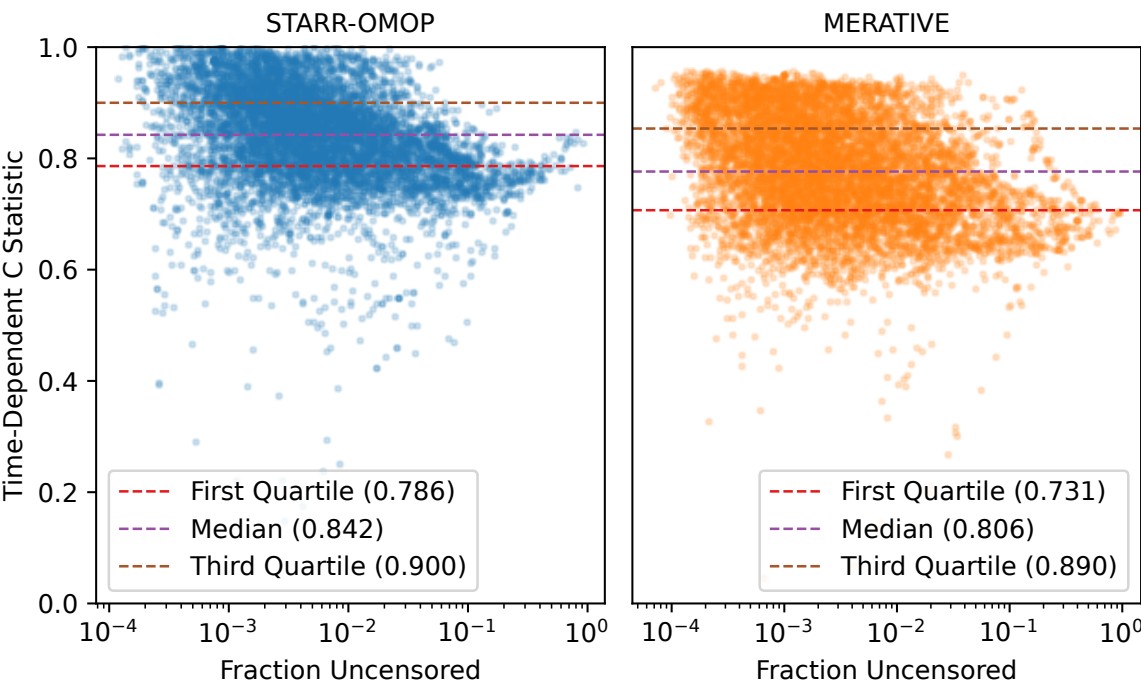

Figure S3: Pretraining task time-dependent C statistic as a function of the fraction of uncensored data. The dotted lines indicate specified quantiles for the C statistic.

### J.11 WEIGHTED EVALUATION CONSIDERING FREQUENCY OF VISITS

In our default evaluation setup, we weight patients equally. This is a common way of evaluating models, especially for the long term TTE predictions we evaluate in our study, but it is also sometimes useful in a clinical setting to consider other weighting strategies.

One other common weighting strategy is to weight by the number of visits so that patients with more clinical interactions are assigned greater weight. In order to understand how our results would change with that weighting strategy, we perform a weighted evaluation in STARR-OMOP where each patient is assigned a weight equal to the number of visits that they have. Table S19 contains the performance of our models and baselines in this setting.

We find that performance numbers do change with this weighting strategy, but that the ranking of methods largely does not.

### J.12 SUBGROUP PERFORMANCE

One commonly accepted necessary condition for a fair model is that it must not have reduced performance within sensitive subgroups (Chang et al., 2022).

In order to evaluate this, we have performed a subgroup analysis comparing the performance in terms of the time-dependent C statistic of MOTOR-Finetune with the strongest baseline RSF within the STARR-OMOP

Table S19: The time-dependent C statistic for all methods on the code-based tasks within STARR-OMOP when patients are weighted by the number of visits that they have. Bolding indicates the best performing model.

| Method | Celiac | HA | Lupus | NAFLD | Cancer | Stroke |
|---|---|---|---|---|---|---|
| Cox PH | 0.642 | 0.634 | 0.706 | 0.680 | 0.688 | 0.699 |
| DeepSurv | 0.653 | 0.730 | 0.715 | 0.731 | 0.721 | 0.746 |
| DSM | 0.655 | 0.730 | 0.730 | 0.746 | 0.720 | 0.744 |
| DeepHit | 0.656 | 0.707 | 0.735 | 0.740 | 0.713 | 0.747 |
| RSF | 0.652 | 0.734 | 0.702 | 0.728 | 0.708 | 0.753 |
| MOTOR-Scratch | 0.648 | 0.711 | 0.772 | 0.762 | 0.668 | 0.750 |
| MOTOR-Probe | 0.757 | 0.802 | 0.768 | 0.800 | 0.767 | **0.805** |
| MOTOR-Finetune | **0.760** | **0.807** | **0.801** | **0.808** | **0.772** | 0.802 |

dataset (MERATIVE lacks ethnicity data and thus cannot be used for a fairness evaluation). Table S20 contains the results of this analysis.

Table S20: The average time-dependent C statistic for RSF, MOTOR-Finetune, and the difference between the two models within important subgroups. The 95% confidence interval for the difference is also included. Bolding indicates the best-performing model. * indicates statistically significant entries at p = 0.05.

| Group | RSF | MOTOR-Finetune | Delta [Confidence Interval] |
|---|---|---|---|
| Female | 0.799 | **0.856** | 0.058 [0.049, 0.067]* |
| Male | 0.790 | **0.845** | 0.055 [0.041, 0.069]* |
| American Indian or Alaska Native | **0.810** | 0.787 | -0.022 [-0.209, 0.171] |
| Asian | 0.793 | **0.853** | 0.059 [0.039, 0.081]* |
| African American | 0.761 | **0.834** | 0.073 [0.006, 0.132]* |
| Native Hawaiian or Other Pacific Islander | 0.841 | **0.914** | 0.073 [-0.044, 0.154] |
| White | 0.788 | **0.850** | 0.062 [0.054, 0.070]* |
| Unknown Race | 0.832 | **0.887** | 0.055 [0.039, 0.073]* |
| Unknown Ethnicity | 0.847 | **0.883** | 0.036 [0.014, 0.065]* |
| Hispanic | 0.794 | **0.857** | 0.063 [0.041, 0.087]* |
| Non-Hispanic | 0.794 | **0.853** | 0.059 [0.052, 0.066]* |

MOTOR-Finetune has better performance within almost every subgroup. The one exception is the very small American Indian or Alaska Native subgroup, and the difference within the margin of error for that group (as shown by the 95% confidence interval containing 0).

## APPENDIX K   NEXT CODE PRETRAINING TASK SETUP

As a baseline, we construct a next code pretraining task in order to better understand the importance of our proposed time-to-event pretraining. Instead of predicting when a future target code occurs, we instead predict whether the code for the next event in the time series is equal to a particular target $k$. This is analogous to the standard autoregressive pretraining task in NLP language modeling. We formalize and implement this next code task as follows.

First, we construct a dictionary of potential target codes to predict. In order to ensure a fair comparison in our context of limited compute, we match the TTE pretraining setup by choosing the 8,192 highest entropy codes as our targets.

Given this fixed dictionary, we can construct a classification task. Each target $k$ is assigned an embedding $E_k$ that is the same width as the internal Transformer representation size. At each event $j$ for every patient $i$, we compute a logit for each target $k$ as the dot product of the final Transformer representation and the embedding $\text{logit}_{ijk} = R_{ij} \cdot E_k$. We can then train our model end-to-end with the standard cross-entropy classification likelihood loss. Let $I_{ijk}$ be an indicator (1 if true, 0 if false) for whether the code for patient $i$ at event $j$'s next event is equal to the target code $k$.

$$L(I|\text{logit}) = \prod_{i,j,k} I_{ijk} \cdot \sigma(\text{logit}_{ij})_k,$$

where $\sigma(\mathbf{z})_l = \frac{e^{z_l}}{\sum_{l'=1}^{n} e^{z_{l'}}}$ is the standard softmax function.

This loss function can then be used to train a model end-to-end using the same setup as for MOTOR (with an Adam optimizer, decreasing learning weight schedule, etc.)

## APPENDIX L    MOTOR MODEL RELEASE

We release a copy of the MOTOR weights trained on STARR-OMOP for external use to aid reproducibility and help others build off this work. The model, and model card, can be found at https://huggingface.co/StanfordShahLab/motor-t-base. Due to the sensitivity of the data involved, we are doing a gated release (Solaiman, 2023) with a mandatory DUA to help prevent inappropriate use. In particular, we explicitly do not permit the use of our released model for providing medical care or to inform medical decision making.

We also take a number of other steps to ensure the safety and privacy of the patients used during pretraining. First, we only train on de-identified, structured data to minimize any identifiable information that could leak into the model. All codes are restricted to a vocabulary defined by public medical ontologies and training data is de-identified to HIPAA standards to remove structured fields that are uniquely identifying, such as patients that are more than 90 years old. Second, all unique textual sequences in our model dictionaries (which are primarily categorical indicators like "Yes" or "No" for lab tests) have been manually verified to contain no PHI. Third, we release our model under a DUA that requires registering the identity of the user of the model and observing a policy where users will make no attempts to re-identify or extract patient level information from the model.

## APPENDIX M    STANDARD DEVIATIONS

Table S21: Estimate and the standard deviation of the difference in time-dependent C statistic between MOTOR-Finetune and the other methods. Higher numbers are considered better for this metric.

| Method | Celiac | Heart Attack | Lupus | NAFLD | Pancretic Cancer | Stroke |
|---|---|---|---|---|---|---|
| | | | STARR-OMOP | | | |
| Cox PH | -0.113 ± 0.013 | -0.126 ± 0.011 | -0.093 ± 0.010 | -0.138 ± 0.004 | -0.072 ± 0.011 | -0.095 ± 0.005 |
| DeepSurv | -0.098 ± 0.012 | -0.063 ± 0.008 | -0.073 ± 0.009 | -0.064 ± 0.010 | -0.054 ± 0.010 | -0.045 ± 0.004 |
| DSM | -0.095 ± 0.010 | -0.059 ± 0.008 | -0.078 ± 0.009 | -0.059 ± 0.003 | -0.056 ± 0.011 | -0.040 ± 0.003 |
| DeepHit | -0.107 ± 0.012 | -0.061 ± 0.009 | -0.055 ± 0.009 | -0.059 ± 0.003 | -0.056 ± 0.011 | -0.042 ± 0.004 |
| RSF | -0.073 ± 0.012 | -0.051 ± 0.008 | -0.076 ± 0.009 | -0.062 ± 0.003 | -0.041 ± 0.010 | -0.035 ± 0.003 |
| MOTOR-Scratch | -0.107 ± 0.012 | -0.092 ± 0.010 | -0.060 ± 0.010 | -0.043 ± 0.003 | -0.088 ± 0.012 | -0.043 ± 0.003 |
| MOTOR-Probe | -0.001 ± 0.005 | -0.003 ± 0.002 | -0.012 ± 0.004 | -0.005 ± 0.002 | -0.001 ± 0.002 | -0.000 ± 0.002 |
| MOTOR-Finetune | 0.000 ± 0.000 | 0.000 ± 0.000 | 0.000 ± 0.000 | 0.000 ± 0.000 | 0.000 ± 0.000 | 0.000 ± 0.000 |
| | | | MERATIVE | | | |
| Cox PH | -0.224 ± 0.005 | -0.048 ± 0.002 | -0.088 ± 0.003 | -0.062 ± 0.003 | -0.207 ± 0.004 | -0.100 ± 0.003 |
| DeepSurv | -0.042 ± 0.003 | -0.017 ± 0.001 | -0.029 ± 0.002 | -0.033 ± 0.003 | -0.033 ± 0.002 | -0.041 ± 0.002 |
| DSM | -0.036 ± 0.003 | -0.017 ± 0.001 | -0.026 ± 0.002 | -0.029 ± 0.003 | -0.030 ± 0.002 | -0.036 ± 0.002 |
| DeepHit | -0.039 ± 0.003 | -0.016 ± 0.001 | -0.029 ± 0.002 | -0.034 ± 0.003 | -0.033 ± 0.002 | -0.041 ± 0.002 |
| RSF | -0.057 ± 0.003 | -0.021 ± 0.001 | -0.033 ± 0.002 | -0.024 ± 0.002 | -0.036 ± 0.002 | -0.048 ± 0.002 |
| MOTOR-Scratch | -0.025 ± 0.002 | -0.010 ± 0.001 | -0.012 ± 0.001 | -0.012 ± 0.002 | -0.013 ± 0.001 | -0.019 ± 0.002 |
| MOTOR-Probe | -0.007 ± 0.002 | -0.003 ± 0.001 | -0.005 ± 0.001 | -0.005 ± 0.001 | -0.009 ± 0.001 | -0.005 ± 0.001 |
| MOTOR-Finetune | 0.000 ± 0.000 | 0.000 ± 0.000 | 0.000 ± 0.000 | 0.000 ± 0.000 | 0.000 ± 0.000 | 0.000 ± 0.000 |

Table S22: 95% confidence intervals for the difference in MOTOR's time-dependent C statistic on the code-based tasks when the pretraining objective is changed. We evaluate both a "next code" baseline as well as our proposed time-to-event objective. ∗ indicates statistically significant entries at p = 0.05.

| Objective | Celiac | Heart Attack | Lupus | NAFLD | Pancretic Cancer | Stroke |
|---|---|---|---|---|---|---|
| Next Code | -0.028 ± 0.009 | -0.025 ± 0.006 | -0.021 ± 0.006 | -0.004 ± 0.002 | -0.006 ± 0.008 | -0.018 ± 0.003 |
| Time-to-Event | 0.000 ± 0.000 | 0.000 ± 0.000 | 0.000 ± 0.000 | 0.000 ± 0.000 | 0.000 ± 0.000 | 0.000 ± 0.000 |

Table S23: Estimate and the standard deviation of the difference in ND calibration between MOTOR-Finetune and other methods. Lower numbers are better for this metric. ∗ indicates statistically significant entries at p = 0.05.

| Method | Celiac | Heart Attack | Lupus | NAFLD | Pancretic Cancer | Stroke |
|---|---|---|---|---|---|---|
| | | | STARR-OMOP | | | |
| Cox PH | 0.075 ± 0.081 | 0.183 ± 0.082 | 0.265 ± 0.097 | 0.077 ± 0.022 | 0.083 ± 0.093 | -0.015 ± 0.023 |
| DeepSurv | -0.051 ± 0.070 | 0.058 ± 0.073 | -0.007 ± 0.064 | 2.868 ± 0.170 | 0.022 ± 0.092 | -0.023 ± 0.020 |
| DSM | 0.449 ± 0.491 | 0.922 ± 1.000 | 0.046 ± 0.146 | -0.031 ± 0.016 | 1.127 ± 2.364 | 0.068 ± 0.142 |
| DeepHit | 1.905 ± 1.132 | 0.203 ± 0.228 | 0.070 ± 0.170 | 0.007 ± 0.029 | 0.453 ± 0.388 | 0.137 ± 0.091 |
| RSF | 0.208 ± 0.102 | 0.194 ± 0.094 | -0.014 ± 0.075 | 0.047 ± 0.024 | 0.286 ± 0.112 | 0.066 ± 0.029 |
| MOTOR-Scratch | 4.878 ± 2.042 | 3.959 ± 2.164 | 23.773 ± 10.433 | 0.034 ± 0.022 | 7.046 ± 4.486 | 0.037 ± 0.040 |
| MOTOR-Probe | -0.047 ± 0.063 | -0.002 ± 0.059 | -0.050 ± 0.054 | -0.024 ± 0.014 | -0.013 ± 0.047 | -0.014 ± 0.021 |
| MOTOR-Finetune | 0.000 ± 0.000 | 0.000 ± 0.000 | 0.000 ± 0.000 | 0.000 ± 0.000 | 0.000 ± 0.000 | 0.000 ± 0.000 |
| | | | MERATIVE | | | |
| Cox PH | -0.074 ± 0.017 | 0.059 ± 0.010 | 0.108 ± 0.018 | 0.190 ± 0.029 | 0.427 ± 0.044 | 0.078 ± 0.009 |
| DeepSurv | -0.061 ± 0.015 | -0.002 ± 0.007 | 19.987 ± 0.753 | 0.036 ± 0.016 | -0.070 ± 0.017 | 3.143 ± 0.147 |
| DSM | -0.048 ± 0.016 | 0.031 ± 0.036 | 0.097 ± 0.025 | 0.081 ± 0.045 | 0.032 ± 0.025 | 0.041 ± 0.031 |
| DeepHit | -0.010 ± 0.025 | 0.088 ± 0.012 | 0.070 ± 0.013 | 0.102 ± 0.070 | -0.014 ± 0.019 | 0.051 ± 0.014 |
| RSF | -0.044 ± 0.015 | 0.039 ± 0.012 | 0.025 ± 0.013 | 0.365 ± 0.038 | 0.005 ± 0.019 | 0.017 ± 0.012 |
| MOTOR-Scratch | -0.096 ± 0.020 | 0.025 ± 0.014 | 0.001 ± 0.009 | 0.051 ± 0.018 | 0.011 ± 0.015 | 0.043 ± 0.009 |
| MOTOR-Probe | -0.094 ± 0.017 | -0.002 ± 0.006 | 0.003 ± 0.010 | 0.001 ± 0.012 | -0.038 ± 0.019 | -0.005 ± 0.005 |
| MOTOR-Finetune | 0.000 ± 0.000 | 0.000 ± 0.000 | 0.000 ± 0.000 | 0.000 ± 0.000 | 0.000 ± 0.000 | 0.000 ± 0.000 |

Table S24: Estimate and the standard deviation of the difference in Harrell C-index between MOTOR-Finetune and the other methods. Higher numbers are considered better for this metric. ∗ indicates statistically significant entries at p = 0.05.

| Method | Celiac | Heart Attack | Lupus | NAFLD | Pancretic Cancer | Stroke |
|---|---|---|---|---|---|---|
| | | | STARR-OMOP | | | |
| Cox PH | $-0.096 \pm 0.014$ | $-0.139 \pm 0.011$ | $-0.088 \pm 0.010$ | $-0.139 \pm 0.005$ | $-0.096 \pm 0.012$ | $-0.099 \pm 0.006$ |
| DeepSurv | $-0.078 \pm 0.011$ | $-0.047 \pm 0.006$ | $-0.064 \pm 0.007$ | $-0.054 \pm 0.003$ | $-0.056 \pm 0.009$ | $-0.037 \pm 0.004$ |
| DSM | $-0.071 \pm 0.010$ | $-0.398 \pm 0.007$ | $-0.072 \pm 0.008$ | $-0.052 \pm 0.003$ | $-0.062 \pm 0.009$ | $-0.038 \pm 0.004$ |
| DeepHit | $-0.107 \pm 0.012$ | $-0.113 \pm 0.012$ | $-0.124 \pm 0.011$ | $-0.186 \pm 0.005$ | $-0.213 \pm 0.018$ | $-0.225 \pm 0.007$ |
| RSF | $-0.054 \pm 0.011$ | $-0.046 \pm 0.006$ | $-0.065 \pm 0.009$ | $-0.069 \pm 0.004$ | $-0.045 \pm 0.010$ | $-0.044 \pm 0.004$ |
| MOTOR-Scratch | $-0.070 \pm 0.010$ | $-0.061 \pm 0.008$ | $-0.058 \pm 0.009$ | $-0.041 \pm 0.002$ | $-0.064 \pm 0.011$ | $-0.040 \pm 0.003$ |
| MOTOR-Probe | $-0.008 \pm 0.004$ | $-0.006 \pm 0.002$ | $-0.023 \pm 0.003$ | $-0.009 \pm 0.001$ | $-0.001 \pm 0.001$ | $-0.003 \pm 0.002$ |
| MOTOR-Finetune | $0.000 \pm 0.000$ | $0.000 \pm 0.000$ | $0.000 \pm 0.000$ | $0.000 \pm 0.000$ | $0.000 \pm 0.000$ | $0.000 \pm 0.000$ |
| | | | MERATIVE | | | |
| Cox PH | $-0.286 \pm 0.003$ | $-0.060 \pm 0.002$ | $-0.104 \pm 0.003$ | $-0.083 \pm 0.004$ | $-0.212 \pm 0.003$ | $-0.118 \pm 0.003$ |
| DeepSurv | $-0.048 \pm 0.003$ | $-0.019 \pm 0.001$ | $-0.033 \pm 0.002$ | $-0.036 \pm 0.003$ | $-0.035 \pm 0.002$ | $-0.044 \pm 0.002$ |
| DSM | $-0.286 \pm 0.003$ | $-0.339 \pm 0.002$ | $-0.350 \pm 0.002$ | $-0.388 \pm 0.003$ | $-0.346 \pm 0.003$ | $-0.305 \pm 0.003$ |
| DeepHit | $-0.066 \pm 0.003$ | $-0.031 \pm 0.002$ | $-0.072 \pm 0.003$ | $-0.164 \pm 0.006$ | $-0.055 \pm 0.002$ | $-0.057 \pm 0.003$ |
| RSF | $-0.066 \pm 0.003$ | $-0.025 \pm 0.001$ | $-0.040 \pm 0.002$ | $-0.035 \pm 0.002$ | $-0.042 \pm 0.002$ | $-0.050 \pm 0.002$ |
| MOTOR-Scratch | $-0.021 \pm 0.002$ | $-0.009 \pm 0.001$ | $-0.008 \pm 0.001$ | $-0.010 \pm 0.002$ | $-0.014 \pm 0.001$ | $-0.014 \pm 0.002$ |
| MOTOR-Probe | $-0.015 \pm 0.002$ | $-0.007 \pm 0.001$ | $-0.011 \pm 0.001$ | $-0.006 \pm 0.001$ | $-0.016 \pm 0.001$ | $-0.011 \pm 0.001$ |
| MOTOR-Finetune | $0.000 \pm 0.000$ | $0.000 \pm 0.000$ | $0.000 \pm 0.000$ | $0.000 \pm 0.000$ | $0.000 \pm 0.000$ | $0.000 \pm 0.000$ |

Table S25: Estimate and the standard deviation of the difference in IBS between MOTOR-Finetune and the other methods. Lower numbers are considered better for this metric. ∗ indicates statistically significant entries at p = 0.05.

| Method | Celiac | Heart Attack | Lupus | NAFLD | Pancretic Cancer | Stroke |
|---|---|---|---|---|---|---|
| | | | STARR-OMOP | | | |
| Cox PH | $0.022 \pm 0.004$ | $0.039 \pm 0.005$ | $0.033 \pm 0.005$ | $0.045 \pm 0.001$ | $0.042 \pm 0.004$ | $0.032 \pm 0.002$ |
| DeepSurv | $0.017 \pm 0.003$ | $0.021 \pm 0.004$ | $0.019 \pm 0.004$ | $0.028 \pm 0.001$ | $0.031 \pm 0.004$ | $0.013 \pm 0.002$ |
| DSM | $0.019 \pm 0.003$ | $0.022 \pm 0.004$ | $0.019 \pm 0.004$ | $0.019 \pm 0.001$ | $0.036 \pm 0.004$ | $0.013 \pm 0.002$ |
| DeepHit | $0.023 \pm 0.004$ | $0.018 \pm 0.004$ | $0.015 \pm 0.004$ | $0.015 \pm 0.001$ | $0.036 \pm 0.005$ | $0.009 \pm 0.002$ |
| RSF | $0.019 \pm 0.003$ | $0.018 \pm 0.004$ | $0.015 \pm 0.004$ | $0.024 \pm 0.001$ | $0.024 \pm 0.003$ | $0.015 \pm 0.002$ |
| MOTOR-Scratch | $0.036 \pm 0.004$ | $0.042 \pm 0.006$ | $0.029 \pm 0.005$ | $0.016 \pm 0.001$ | $0.053 \pm 0.007$ | $0.017 \pm 0.002$ |
| MOTOR-Probe | $-0.001 \pm 0.002$ | $0.000 \pm 0.001$ | $0.009 \pm 0.002$ | $0.000 \pm 0.001$ | $0.002 \pm 0.001$ | $-0.005 \pm 0.001$ |
| MOTOR-Finetune | $0.000 \pm 0.000$ | $0.000 \pm 0.000$ | $0.000 \pm 0.000$ | $0.000 \pm 0.000$ | $0.000 \pm 0.000$ | $0.000 \pm 0.000$ |
| | | | MERATIVE | | | |
| Cox PH | $0.029 \pm 0.001$ | $0.015 \pm 0.001$ | $0.026 \pm 0.001$ | $0.034 \pm 0.001$ | $0.038 \pm 0.001$ | $0.019 \pm 0.001$ |
| DeepSurv | $0.011 \pm 0.001$ | $0.007 \pm 0.000$ | $0.033 \pm 0.001$ | $0.017 \pm 0.001$ | $0.009 \pm 0.001$ | $0.038 \pm 0.001$ |
| DSM | $0.010 \pm 0.001$ | $0.008 \pm 0.000$ | $0.009 \pm 0.001$ | $0.015 \pm 0.001$ | $0.009 \pm 0.001$ | $0.010 \pm 0.001$ |
| DeepHit | $0.013 \pm 0.001$ | $0.009 \pm 0.001$ | $0.011 \pm 0.001$ | $0.020 \pm 0.001$ | $0.012 \pm 0.001$ | $0.009 \pm 0.001$ |
| RSF | $0.014 \pm 0.001$ | $0.008 \pm 0.001$ | $0.010 \pm 0.001$ | $0.016 \pm 0.001$ | $0.011 \pm 0.001$ | $0.012 \pm 0.001$ |
| MOTOR-Scratch | $0.005 \pm 0.000$ | $0.003 \pm 0.000$ | $0.002 \pm 0.000$ | $0.005 \pm 0.001$ | $0.003 \pm 0.000$ | $0.005 \pm 0.000$ |
| MOTOR-Probe | $0.005 \pm 0.000$ | $0.002 \pm 0.000$ | $0.003 \pm 0.000$ | $0.003 \pm 0.001$ | $0.004 \pm 0.001$ | $0.002 \pm 0.000$ |
| MOTOR-Finetune | $0.000 \pm 0.000$ | $0.000 \pm 0.000$ | $0.000 \pm 0.000$ | $0.000 \pm 0.000$ | $0.000 \pm 0.000$ | $0.000 \pm 0.000$ |

## APPENDIX N   CODE SELECTION FOR PRETRAINING

Due to compute limits, it is difficult to pretrain using every code as a pretraining task. Thus, we have to select a subset of codes for pretraining (8,192 for our main experiments). In order to perform this selection, we use a relatively simple heuristic designed to be equivalent to frequency based ranking (i.e. taking the most frequently used codes in the dataset), but taking into account the ontology relationships between the possible pretraining tasks.

In the absence of ontology relationships, frequency based ranking is equivalent to ranking by Shannon entropy (Shannon, 1948). Therefore, to have a ranking criterion that is similar to frequency based selection, but taking into account the ontological relationship between codes, we use the conditional Shannon entropy formula, conditioning on the presence or absence of the parents of a particular code.

Let $C$ be a boolean random variable indicating the presence or absence of a particular code in a patient timeline and let $O$ be a boolean random variable indicating the presence or absence of the parents of that code.

The conditional entropy formula is then:

$$H(C|O) = - \sum_{c \in \mathcal{C}, o \in \mathcal{O}} p(o, c) \log \frac{p(o, c)}{p(o)}$$

$C$ and $O$ are both boolean and can only take the values True (T) or False (F), so we can expand out the above formula.

$$
\begin{aligned}
H(C|O) = &- p(O = F, C = F) \log \frac{p(O = F, C = F)}{p(O = F)} + \\
&- p(O = F, C = T) \log \frac{p(O = F, C = T)}{p(O = F)} + \\
&- p(O = T, C = F) \log \frac{p(O = T, C = F)}{p(O = T)} + \\
&- p(O = T, C = T) \log \frac{p(O = T, C = T)}{p(O = T)}
\end{aligned}
\tag{6}
$$

We can simplify this equation by taking advantage of the fact that these are ontologies where a patient must have higher level codes if they have a lower level code, which implies that $p(O = F, pfC = T) = 0$ and $p(O = F, C = F) = p(O = F)$. Thus the formula can be simplified to:

$$
\begin{aligned}
H(C|O) = &- p(O = T, C = F) \log \frac{p(O = T, C = F)}{p(O = T)} + \\
&- p(O = T, C = T) \log \frac{p(O = T, C = T)}{p(O = T)}
\end{aligned}
\tag{7}
$$

This formula can then be evaluated by plugging in $p(O = T, C = F)$, $p(O = T, C = T)$, and $p(O = T)$, both of which can be estimated by counting the frequency of codes (conditioned on the presence/absence of their parents) in the training data. Finally, we select a set of 8,192 codes for pretraining by taking the top 8,192 codes that have the highest entropy.

