# OpenReview forum: "MOTOR: A Time-to-Event Foundation Model For Structured Medical Records"
_ICLR.cc/2024/Conference — ICLR 2024 spotlight_

### Official Review · Reviewer_epjm · 2023-10-30

**Soundness:** 3 good
**Presentation:** 4 excellent
**Contribution:** 3 good
**Rating:** 8
**Confidence:** 4

**Summary:**

The authors of this paper propose a survival analysis foundation model from ICD codes event sequences. Their proposed pre-training method is to leverage the specific nature of the data being itself a sequence of events to perform a multitask dynamic survival analysis training by predicting the time-to-event of 8192 higher-entropy codes in their data (excluding the downstream task ones). Their model is a transformer backbone with heads for each pre-training task returning piecewise-continuous hazard. They compare their model to a popular survival analysis baseline as well as next-event prediction pre-training on 6 survival tasks across two datasets. They also evaluate their foundation model's robustness to time shifts and external datasets.

**Strengths:**

Overall I find the paper to be of high quality. The contribution is significant, the experiments thought carefully and the paper very clear and complete.


### Novelty
To the best of my knowledge, this is the first paper proposing a foundation mode for survival analysis. This is an important contribution, as deep learning applications in the field, due to the relative smallness of available datasets, are less common and with rather small-scale models.

I find it very creative to use EHR data for survival analysis as it is usually only used in the context of (fixed horizon) early event prediction (e.g. Tomasev et al 2019, Hyland et al 2020).


### Clarity

The paper is very easy to follow because very well organized. All necessary details are present in the main text or relevant appendices.


### Experiments

The authors did a great job at comparing to many popular survival baselines as their tasks were not commonly used ones. I particularly appreciate that they made sure to remove the codes used in their downstream tasks from the pre-training as an extra security measure against leakages. Finally, the scope of experiments they considered, especially the validation of their model on external data, really strengthened their work.

The overall performance improvement is notable. in particular in a field where performance gains are usually of very low magnitude.

**Weaknesses:**

### The specificity of the task to ICD codes data
If using the fact that ICD code sequences are sequences of events rather than observation to directly perform large-scale multitask dynamic survival analysis on them, is great for this type of data, it is also very limiting to this unique type of data. Indeed, as mentioned by the authors, survival analysis is useful in a variety of domains ranging from cancer research to finance. However,  in such domains, to my knowledge, data cannot be formalized as a sequence of events, hence the proposed pretraining task could not be expanded to other time-to-event models.

### The lack of experiments in the dynamic setting
The authors pre-train their model in the so-called "dynamic survival analysis" setting, however, they only evaluate it in a static setting. It would have been a great addition to also have dynamic tasks. The same baselines can be considered with a landmarking approach.

### The drop in performance for the Native American sub-group

The attention of the authors to ethical concerns is clearly above standard. As part of it, they perform a sub-group analysis per ethnicity. Compared to the RSF model which is quite stable across groups, their model exhibits a significant drop in performance among Native Americans (~5% lower than the closest group). However, the authors have the following statement: "We find that with one statistically insignificant exception, MOTOR-Finetune does not reduce the performance within sensitive groups". I have a hard time believing that such a drop is "statistically insignificant". I believe the authors should clearly state this limitation.

**Questions:**

- How does the author handle events that can occur multiple times? Do they predict the time of the next event instead?

---

> ### Author Response · Authors · 2023-11-18
> **Official Response to Reviewer epjm**
>
> Thank you for the review and helpful feedback. The following contains answers to your concerns and questions:
>
> > The specificity of the task to ICD codes data
>
> Our method is applicable to datasets that have timestamped event information. Such datasets are not restricted to medicine and do exist in enterprise settings. For example, user activity datasets that track user interactions within a particular company / software app (such as [1] and [2]) have similar streams of timestamped event data that may be suitable for our method.  One challenge is that these timestamped, event datasets are difficult for researchers to obtain at scale because companies generally view them as high-value, proprietary data.
>
> Medical records offer the advantage of being available at the scale necessary for training foundation models and are generally more accessible to researchers (e.g., MIMIC-IV, MERATIVE). However, we are hopeful that future work will be able to reproduce our results on other sources of timestamped event datasets.
>
>
> 1. How Jing and Alexander J. Smola. 2017. Neural Survival Recommender. In Proceedings of the Tenth ACM International Conference on Web Search and Data Mining (WSDM '17). Association for Computing Machinery, New York, NY, USA, 515–524. https://doi.org/10.1145/3018661.3018719
>
> 2. Á. Periáñez, A. Saas, A. Guitart and C. Magne, "Churn Prediction in Mobile Social Games: Towards a Complete Assessment Using Survival Ensembles," 2016 IEEE International Conference on Data Science and Advanced Analytics (DSAA), Montreal, QC, Canada, 2016, pp. 564-573, doi: 10.1109/DSAA.2016.84.
>
> > The lack of experiments in the dynamic setting
>
>
> We agree that more exploration of dynamic vs static survival analysis settings is an important research direction and that the lack of dynamic setting evaluations is a limitation. We hope to explore these types of evaluation settings in future work.
>
> > The drop in performance for the Native American sub-group
>
> We apologize for not providing sufficient details to make a convincing case. The issue is that the Native American subgroup is very small in EHR-OMOP, only 0.3% of the total patients fall into that category, so our estimates for that group are very wide. We have explicitly added confidence intervals to Appendix J.12 (table reproduced below) to better illustrate the issue. Note how the confidence interval for the Native American sub-group contains 0, and is thus not statistically significant.
>
> The table below contains the average time-dependent C statistic for RSF, MOTOR-Finetune, and the difference between the two models within important subgroups. The 95\% confidence interval for the difference is also included. Bolding indicates the best-performing model. $\ast$ indicates statistically significant entries at p = 0.05.
>
> |Group|RSF|MOTOR-Finetune|Delta [Confidence Interval]|
> |-|-|-|-|
> |Female|0.799|**0.856**|0.058 [0.049, 0.067]\*|
> |Male|0.790|**0.845**|0.055 [0.041, 0.069]\*|
> |-|-|-|-|
> |American Indian or Alaska Native|**0.810**|0.787|-0.022 [-0.209, 0.171]|
> |Asian|0.793|**0.853**|0.059 [0.039, 0.081]\*|
> |African American|0.761|**0.834**|0.073 [0.006, 0.132]\*|
> |Native Hawaiian or Other Pacific Islander|0.841|**0.914**|0.073 [-0.044, 0.154]|
> |White|0.788|**0.850**|0.062 [0.054, 0.070]\*|
> |Unknown Race|0.832|**0.887**|0.055 [0.039, 0.073]\*|
> |-|-|-|-|
> |Unknown Ethnicity|0.847|**0.883**|0.036 [0.014, 0.065]\*|
> |Hispanic|0.794|**0.857**|0.063 [0.041, 0.087]\*|
> |Non-Hispanic|0.794|**0.853**|0.059 [0.052, 0.066]\*|
>
>
>
> The differences for the Native Hawaiin or Other Pacific Islander racial category are also not statistically significant, which reflects how only 1% of patients in EHR-OMOP belong to that racial group.
>
> We agree that performance in these groups is important, so we have added an explicit disclaimer in our model card that our model is not tested well on those two subgroups. We hope that our model release will help address this problem by allowing other researchers with access to those populations to validate our model for those groups.
>
> > How does the author handle events that can occur multiple times? Do they predict the time of the next event instead?
>
> We predict the time until the next event for events that can occur multiple times. We have modified the relevant sentence in the methods section to make this more clear.

---

> > ### Comment · Reviewer_epjm · 2023-11-18
> > **Answer to authors**
> >
> > Thank you for the clarifications !
> >
> > I'll maintain my score and I have no more questions.

---

### Official Review · Reviewer_ZCsF · 2023-11-01

**Soundness:** 3 good
**Presentation:** 4 excellent
**Contribution:** 3 good
**Rating:** 8
**Confidence:** 3

**Summary:**

This paper proposes MOTOR (Many Outcome Time Oriented Representations), a new time-to-event (TTE) foundation model trained on 55M patients with 9B clinical events. Instead of using classification to predict medical events with the assumption of fixed time horizons, MOTOR adopts TTE modeling for its better principled way of predicting events with time and the ability to handle censored observations. Experimental results on three EHR databases show that MOTOR outperforms existing TTE models and is more robust to temporal distributional shifts, suggesting the potential use of pre-training models in TTE modeling.

**Strengths:**

- The paper is well-written and easy to follow.
- Interesting to observe the potential of time-to-event modeling trained on a large amount of data.
- The experiments are thorough and convincing (especially text-based target tasks are interesting to include)
- The proposed method outperforms previous survival time-to-event methods.

**Weaknesses:**

- Limited ablation on the amount of pretraining data (e.g., 1%, 10%). Although the authors have made a significant contribution by releasing the model, providing more details on the amount of data necessary to achieve the current performance would be highly beneficial. Time-to-event modeling holds importance not only in medical settings but also in various other domains.
- Figure 1 could be explained or illustrated in more detail. For example, it is hard to understand what each survival curve means for each timestep in pretraining tasks.

**Questions:**

- How much time did it take to pretrain the model?
- What is the source of pretraining data?
- Can you elaborate more on the 8,192 code selection process?
- Is the patient data at the admission level or concatenated across admissions?

---

> ### Author Response · Authors · 2023-11-18
> **Official Response to Reviewer ZCsF**
>
> Thank you for the review and helpful feedback. We address your concerns and questions below:
>
> > Limited ablation on the amount of pretraining data (e.g., 1%, 10%). Although the authors have made a significant contribution by releasing the model, providing more details on the amount of data necessary to achieve the current performance would be highly beneficial. Time-to-event modeling holds importance not only in medical settings but also in various other domains.
>
> We agree that a deeper analysis of TTE pretraining, including more detailed analyses of pretraining sizes, is an exciting direction for further exploration. In the presence of limited compute budgets, our current pretraining sweep shown in Figure 4 used 5%, 10%, 25%, and 100% of the input data. Our pretraining ablations are able to highlight 2 points:
>
> (1) In EHR data (EHR-OMOP), we observed that pre-training provides larger gains when using more data, suggesting potentially further gains beyond the 2.5M patients that comprised our entire EHR patient pre-training set.
>
> (2) In claims data (MERATIVE), we observed some benefits from pre-training beyond our smallest sample (2.74M patients), but overall total benefit was 89% less than the gain from EHR data.
>
> These findings provide some guidance on how much data practitioners need in that small EHR datasets are insufficient to pretrain MOTOR effectively and pretraining MOTOR on very large claims datasets does not provide significant additional predictive value.
>
> > Figure 1 could be explained or illustrated in more detail. For example, it is hard to understand what each survival curve means for each timestep in pretraining tasks.
>
> Thank you for pointing out the need for Figure 1 needing more explanation and better element labels. For the survival plots, where the x axis is time and the y axis is the probability, each time step adds new information that can update the estimate of y in the survival curve.
>
> We have updated the figure capture to explain this intuition and better label figure elements.
>
> > How much time did it take to pretrain the model?
>
> Pretraining on EHR-OMOP takes 111 V100 GPU hours. Pretraining on MERATIVE takes 330 V100 GPU hours. We have added this detail to the compute section in Appendix C.
>
> > What is the source of pretraining data?
>
> We pretrain on two datasets, EHR-OMOP and MERATIVE. EHR-OMOP is a private dataset provided by an academic medical center. MERATIVE is commercially available and our institution purchased a license.
>
> > Can you elaborate more on the 8,192 code selection process?
>
> We have added Appendix O with a more comprehensive description of our code selection process. We have reproduced a summarized version of that new section here. The core idea is to do code selection using a simple heuristic that, as closely as possible, matches a standard frequency-based selection technique (i.e., including the top-k most common codes for pretraining) but accounts for the ontology relationships between codes. In the absence of ontology relationships, ranking using frequency is equivalent to ranking by Shannon entropy. Therefore, to have a ranking criteria that is similar to frequency-based selection, but taking into account ontologies, we use the conditional Shannon entropy formula, conditioning on the presence or absence of the parents of a particular code.
>
> Let $C$ be a boolean random variable indicating the presence or absence of a particular code in a patient timeline and let $O$ be a boolean random variable indicating the presence or absence of the parents of that code.
>
> The conditional entropy formula is:
>
> $$H(C|O)\ = -\sum_{c\in\mathcal C, o\in\mathcal O}p(o,c)\log \frac {p(o, c)} {p(o)}$$
>
> This simplifies to:
>
> \begin{equation}
> H(C|O) =    -p(O = T,C = F)\log \frac{p(O = T, C = F)} {p(O = T)} +  -p(O = T,C = T)\log \frac{p(O = T, C = T)} {p(O = T)}
> \end{equation}
>
> This formula can then be evaluated by plugging in $p(O = T, C = F)$, $p(O = T, C = T)$, and $p(O = T)$, all of which can be estimated by counting the frequency of codes (conditioned on the presence / absence of their parents) in the training data. The highest entropy codes according to this formula are then selected.
>
>
> > Is the patient data at the admission level or concatenated across admissions?
>
> The patient data is concatenated across admissions.

---

> ### Comment · Reviewer_ZCsF · 2023-11-23
> **Answer to authors**
>
> Thank you for the clarifications!

---

### Official Review · Reviewer_MVk7 · 2023-11-01

**Soundness:** 3 good
**Presentation:** 3 good
**Contribution:** 4 excellent
**Rating:** 6
**Confidence:** 3

**Summary:**

The authors propose a Transformer model uses a self-supervised TTE objective defined using structured EHR data.

**Strengths:**

1) Paper is well-written and results are interesting. Authors also cover a number of pre-training tasks including code-based and text-based
2) Authors compare with a number of relevant baselines, and clearly show the impact of their proposed approach
3) The authors test robustness and importance of specific pre-training tasks, and show that their model is relatively robust to different datasets

**Weaknesses:**

1) It is not clear how the authors handle missing information in their analyses. How are missing values in the time-series data represented?
2) While the authors explain the subsampling process in the Appendix, this is critical and forms the basis of the pre-training approach. Expanding on the rationale/method for this seems important

**Questions:**

How are missing values in the time-series data represented? Are specific transformer base models used?

---

> ### Author Response · Authors · 2023-11-18
> **Official Response to Reviewer MVk7**
>
> Thank you for the review and helpful feedback. We have compiled point-by-point answers to your questions and concerns below.
>
> > It is not clear how the authors handle missing information in their analyses. How are missing values in the time-series data represented?
>
> We acknowledge that missing data (particularly informative missingness) is an important aspect of modeling EHR data. As noted in Appendix F, we follow BEHRT[1]’s general Transformer approach, which attempts to handle missingness using a sequence formulation, where patient information is received one event at a time, events with missing data are not generated. Because event times are provided as a feature to a deep neural network architecture that can process complex feature interactions, in principle the model is capable of learning informative patterns of missingness when certain events are systematically absent.
>
> We have updated Appendix F to include the above information.
>
> 1. Li, Y., Rao, S., Solares, J.R.A. et al. BEHRT: Transformer for Electronic Health Records. Sci Rep 10, 7155 (2020). https://doi.org/10.1038/s41598-020-62922-y
>
>
> > While the authors explain the subsampling process in the Appendix, this is critical and forms the basis of the pre-training approach. Expanding on the rationale/method for this seems important.
>
>
> Thank you for the suggestion to expand on this matter. The subsampling approach noted in Section 4 is not used in any way as part of our pre-training process. Subsampling is only performed when creating target task datasets for evaluation purposes, given the scale of our evaluation datasets (i.e., containing up to tens of millions of patients), which makes it infeasible to evaluate more computationally expensive baseline methods like RSF on the entire dataset. Section 4 and Appendix D elaborate on this motivation.
> While this subsampling does not impact the relative ranking and comparisons of methods in our manuscript, it does induce a non-uniform scaling of the hazard rates. Since calculating the unbiased hazard rate is often useful, Appendix D also formally outlines the exact hazard rate scaling induced by our evaluation subsampling.
>
> > Are specific transformer base models used?
>
> Sorry for the confusion. We specified this in the appendix, but not in the main body of the paper. We use a local attention transformer base model, with the exact implementation details matching the local attention base model described in [1]. We have since moved this detail and reference into the main body of the paper.
>
> 1. Aurko Roy, Mohammad Saffar, Ashish Vaswani, and David Grangier. Efficient content-based sparse attention with routing transformers. Transactions of the Association for Computational Linguistics, 9:53–68, 2021. doi: 10.1162/tacl_a_00353. URL https://aclanthology.org/2021.tacl-1.4

---

### Official Review · Reviewer_13Zf · 2023-11-01

**Soundness:** 3 good
**Presentation:** 3 good
**Contribution:** 4 excellent
**Rating:** 8
**Confidence:** 4

**Summary:**

In this work, the authors propose a time-to-event (TTE) foundation model for EHR data. The proposed model is retrained on a 55M patient record data and 8192 tasks. The model is evaluated on 19 tasks across 3 patient databases and achieves superior performances.

**Strengths:**

This work addresses an important task in medical time-to-event prediction domain and the results are promising. The trained model and code is publicly available. The experiment results and analysis are comprehensive. Here are some minor comments:

1. What are the x and y axis in Figure 1 - pretraining tasks? Are they hazard ratio curves?

2. How the six code-based tasks are selected out of 8192 tasks? Please provide more justifications for the selection.

3. How the time-to-event task is evaluated on the MIMIC dataset? Is it predicting the diagnosis code at each ICU admission?

4. The performance table seems lack of standard deviations.

5. How much resources are needed to fine-tune or inference using the pretrained model?

**Weaknesses:**

Here are some minor comments:

1. What are the x and y axis in Figure 1 - pretraining tasks? Are they hazard ratio curves?

2. How the six code-based tasks are selected out of 8192 tasks? Please provide more justifications for the selection.

3. How is the time-to-event task evaluated on the MIMIC dataset? Is the goal to predict the diagnosis code at each ICU admission?

4. There are no standard deviations in performance tables.

5. How many resources are needed to fine-tune or infer using the pretrained model?

**Questions:**

Please address the weaknesses above.

---

> ### Author Response · Authors · 2023-11-18
> **Official Response to Reviewer 13Zf**
>
> Thank you for the thoughtful review and helpful feedback. We provide answers to your questions below:
>
> > 1. What are the x and y axis in Figure 1 - pretraining tasks? Are they hazard ratio curves?
>
>
> We acknowledge this figure was confusing in our original submission. These are survival curves, with the x axis as time and the y axis as the probability that the event occurs after x time. We have updated the figure to more clearly label the axes and other figure elements.
>
> > 2. How the six code-based tasks are selected out of 8192 tasks?
>
> Our six target tasks aim to encompass a variety of acute and chronic health conditions, affecting various body systems and occurring at different prevalence rates. We focus on situations where a clinical case has been made for the utility of having a prediction for that condition. Stroke and heart attack prediction have the most mature clinical utility, as risk predictions for those conditions are key components for the widely deployed ASCVD risk scoring systems recommended by clinical guidelines that are currently used to drive statin and hypertensive therapy for millions of patients [1]. The other conditions are not as mature (in that a prediction model for them is not part of current guidelines), but do appear to have promise for future clinical applications. For example, pancreatic cancer is particularly promising as a prediction target because early detection is critical for high patient survival and current human physician diagnosis frequently occurs way too late to save many patients [2]. Likewise, there are similar arguments for the clinical utility of models for predicting future diagnoses of NAFLD [3], Celiac Disease [4] and Lupus [5].
>
> 1. Wong ND, Budoff MJ, Ferdinand K, Graham IM, Michos ED, Reddy T, Shapiro MD, Toth PP. Atherosclerotic cardiovascular disease risk assessment: An American Society for Preventive Cardiology clinical practice statement. Am J Prev Cardiol. 2022 Mar 15;10:100335. doi: 10.1016/j.ajpc.2022.100335. PMID: 35342890; PMCID: PMC8943256.
>
> 2. Lee HA, Chen KW, Hsu CY. Prediction Model for Pancreatic Cancer-A Population-Based Study from NHIRD. Cancers (Basel). 2022 Feb 10;14(4):882. doi: 10.3390/cancers14040882. PMID: 35205630; PMCID: PMC8870511.
>
> 3. Razmpour, F., Daryabeygi-Khotbehsara, R., Soleimani, D. et al. Application of machine learning in predicting non-alcoholic fatty liver disease using anthropometric and body composition indices. Sci Rep 13, 4942 (2023). https://doi.org/10.1038/s41598-023-32129-y
>
> 4. Hujoel IA, Murphree DH Jr, Van Dyke CT, Choung RS, Sharma A, Murray JA, Rubio-Tapia A. Machine Learning in Detection of Undiagnosed Celiac Disease. Clin Gastroenterol Hepatol. 2018 Aug;16(8):1354-1355.e1. doi: 10.1016/j.cgh.2017.12.022. Epub 2017 Dec 16. PMID: 29253540; PMCID: PMC6004230.
>
> 5. Kingsmore, Kathryn M.; Lipsky, Peter E.. Recent advances in the use of machine learning and artificial intelligence to improve diagnosis, predict flares, and enrich clinical trials in lupus. Current Opinion in Rheumatology 34(6):p 374-381, November 2022. | DOI: 10.1097/BOR.0000000000000902
>
> We have added the above citations and rationale of task choice to the Experiments section.
>
> > 3. How the time-to-event task is evaluated on the MIMIC dataset? Is it predicting the diagnosis code at each ICU admission?
>
> We take advantage of the longitudinal non-ICU data provided by MIMIC-IV, which allows us to use the same task setup as within MERATIVE and EHR-OMOP, aka predicting the time until the first diagnosis of a condition from the end of a given visit. We have updated Appendix J.7 (which describes the MIMIC experiments) to make this more clear.
>
>
> > 4. The performance table seems lack of standard deviations.
>
>
> Thank you for the suggestion to include standard deviations. We now include them in section N of the appendix.
>
> > 5. How much resources are needed to fine-tune or inference using the pretrained model?
>
> Our timing experiments (which are detailed in Appendix J.10) show that our model has a throughput of 147 patients per second on EHR-OMOP when performing inference using a single NVIDIA V100. We have added that throughput number to Appendix J.10.

---

> > ### Comment · Reviewer_13Zf · 2023-11-20
> >
> > Thanks to the authors for their comprehensive replies. I believe this is a strong paper to accept.

---

### Author Response · Authors · 2023-11-18
**General Response**

We would like to thank all of the reviewers for their overall positive response to our work, helpful feedback, and suggestions for improvement. Given the importance of time-to-event modeling with structured medical records, we appreciate the opportunity to make our contributions more clear.

In light of all reviewer feedback, we have updated our manuscript with the following major changes:
-  We have revamped Figure 1 with clearer element labeling.
-  We have added information on the compute resources used for experiments, including details on model throughput and pretraining time.
-  We have added additional statistical uncertainty estimates for various tables, especially the subgroup analyses.
- We have written a new appendix section (Appendix O) with a detailed description of how pre-training tasks were selected.

---

### Meta-Review · Area_Chair_oodQ · 2023-12-06

**Metareview:**

This paper introduces, MOTOR, a self-supervised foundation model, which is pre-trained on timestamped sequences from electronic health records and health insurance claims, for time-to-event (TTE) predictions. It excels in estimating event occurrence probabilities, outperforming state-of-the-art models in 19 tasks across 3 patient databases. MOTOR demonstrates improved label efficiency and robustness, showcasing significant advancements in medical TTE predictions. The authors well-addressed the reviewers' comments. This paper is well-written and easy to follow and the proposed method outperforms the methods in the literature. Overall, this is a great paper.

**Justification For Why Not Higher Score:**

The reviewers were not enthusiastic enough for this paper to be an oral one.

**Justification For Why Not Lower Score:**

I think this is a great paper that deserves a better showcase than a regular poster.

---

### Decision · Program_Chairs · 2024-01-16

Accept (spotlight)